

# Refined glacial lake extraction in high Asia region by Deep Neural Network and Superpixel-based Conditional Random Field

Yungang Cao[1], Xueqin Bai[1], Meng Pan[2], Ruodan Lei[1], Puying Du[1]

[1]Faculty of Geosciences and Environmental Engineering, Southwest Jiaotong University, Chengdu,611756, China
[2]College of Civil Engineering, Shangqiu Institute of Technology, Shangqiu,476000, China

*Correspondence to*: *Yungang Cao* (yungang@swjtu.cn)

**Abstract**: Remote sensing extraction of glacial lakes is an effective way of monitoring water body

distribution and outburst disasters. At present, the lack of glacial lake datasets and the edge recognition problem of semantic segmentation networks lead to poor accuracy and inaccurate outlines of glacial lakes. Therefore, this study constructed a high-resolution dataset containing seven types of glacial lakes and proposed a refined glacial lake extraction method, which combines the LinkNet50 Network for rough extraction and Simple Linear Iterative Clustering (SLIC)-Dense Conditional Random Field (DenseCRF)

for optimization. The results show that: 1) With Google Earth images of 0.52 m resolution in the study area, the Recall, Precision, F1 Score, and IoU of glacial lake extraction based on the proposed method are 96.52%, 92.49%, 94.46%, and 90.69%, respectively. 2) With the Google Earth images of 2.11 m resolution in the Qomolangma National Nature Reserve, 2300 glacial lakes with a total area of about 65.17 km$^2$ were detected by the proposed method. The area of the minimum glacial lake that can be

extracted is about 160 m$^2$ (6×6 pixels). This method has advantages in small glacial lake extraction and refined outline detection, which can be applied to extracting glacial lakes in the high Asia region with high-resolution images.

## 1 Introduction

Glacial lakes are natural water bodies mainly supplied by glacier meltwater or formed by water

accumulation in moraine ridge depressions and are densely distributed in high Asia (Yao et al. 2017). Glacial lakes have a strong relationship to ongoing climate warming (Pandey et al. 2021). On the one hand, in the global warming environment, increased glacier runoff and glacial lake outburst floods caused by melting glaciers threaten the lives and property of surrounding residents (Song et al. 2016; Begam et al. 2018). On the other hand, as a product of global warming and glacier melting, glacial lakes are one of

the sensitive indicators reflecting global changes (Lei et al. 2014; Qiu et al. 2019; Zhou et al. 2019). Besides, many glacial lakes are small in size and unevenly distributed (Yang et al. 2018). Small glacial lakes are more active and sensitive to climate change (Sakai et al. 2015; Zhang et al. 2015). Therefore, accurate monitoring of glacial lakes is essential to studies on global climate change, water resource distribution, and disaster warnings.

For the extraction methods of glacial lakes, there are mainly manual digitization methods, semi-automatic methods, and automatic methods. First, the manual digitization method has achieved good



results. Still, it costs lots of time and vigor, which is challenging to meet the needs for large-scale glacial lake identification. For the semi-automatic extraction of glacial lakes, current studies are mainly based on water body indices (Li et al. 2020) and machine learning. In 2015, Jain et al. (2015) used the Support Vector Machine (SVM) to detect glacial lakes in Bhutan, Himalayas. In 2018, Veh et al. (2018) trained a Random Forest (RF) classifier based on Landsat data and detected glacial lake outbursts through change detection technology. These semi-automatic extraction methods are widely used, but the operation is manually dependent and regionally restrictive, limiting the promotion and application on the global/hemispheric scale. For example, the accuracy of the results obtained by machine learning methods (SVM, RF et al.) depends mainly on the selection of prior knowledge and the reliability of training samples.

For automatic extraction methods, there are mainly image segmentation methods (Zhang et al. 2018) and edge detection algorithms (Cordeiro et al. 2021). These methods establish fixed models or rules and then execute them automatically, completing extraction work without manual intervention. The canny edge detection algorithm is one of the most classical and advanced image edge detection algorithms (Chen, 2021). In the glacial lake extraction, although the threshold could be automatically specified, the extracted glacial lake edge was not complete. If a uniform threshold was applied to the whole image, some small glacial lakes would be missed. Threshold and Simplified C-V (TSCV) based on image segmentation technology has a better effect (Zhao et al. 2018), which could overcome the impact of spectral heterogeneity. However, the calculation procedure of this method is complicated and the robustness of the algorithm is not ideal. It is only applicable to Landsat images and has limitations for the identification of fine lake edges and small glacial lakes. For high-resolution images, this method is still lacking in testing large-scale glacial lake extraction.

Except for the traditional automatic image segmentation methods, with the development of computer vision technology, some image semantic segmentation networks have been successfully applied in water body recognition (Chen et al. 2018; Talal et all. 2018; Wang et al. 2019; Wang et al. 2022a). Based on PlanetScope Imagery, Qayyum et al. (2020) used the pre-trained EfficintNet as the backbone of the U-Net to map glacial lakes, which achieved a better result in high-resolution glacial lakes extraction. But given that the area of the glacial lake is much smaller compared with the background. Skip connection structure will transfer a large amount of redundant background information from the low level to the high level, reducing the utilization efficiency of low level features. He et al. (2021) added a space attention mechanism into the skip connection of U-Net to focus on glacial lakes. With NDWI as the spatial attention, NAU-Net guided the network to pay more attention to the glacial lake information of low-level features and solved the problem of the area difference between positive and negative samples (Wang et al. 2022b). However, for high-resolution Google Earth images, there are more problems with complex spectral and texture features that lead to the large intraclass variance of glacial lakes. Therefore, based on high spatial resolution data, Wang et al. (2020) extracted lakes on the Tibetan Plateau with a more complex network(MSLWENet). Although the texture of the water body was complex, resulting in more noise in the segmentation, the study showed that the deeper network achieved better performance than U-Net, DeepLab V3+ (Li et al. 2019), and et al.



For end-to-end semantic segmentation networks, the network is vulnerable to negative samples because glacial lakes have small areas, and part of the spatial information is difficult to recover during up-sampling (Song et al. 2019). Besides, high-resolution images provide not only rich spectral information of glacial lakes but also contain a lot of noise information and a deep network is needed.

Considering the characteristics of high-resolution images and the limitations of semantic segmentation networks, this study proposed an automatic method for the refined glacial lake extraction. The main contributions of this study are as follows:

(1) A glacial lake dataset with abundant glacial lake types and sufficient samples was constructed in this study.

(2) To alleviate the negative impact of unbalanced positive and negative samples on the network extraction for glacial lake features, the loss function of the network with Resnet50 as the backbone was modified.

(3) Simple Linear Iterative Clustering (SLIC) and DenseCRF were combined for post-processing to reduce the noise of segmentation results and optimize glacial lake outlines.

**2 Study area and data**

**2.1 Study area**

The study area is undertaken in the Mount Qomolangma area (27°08'09" N~29°19'14"N, 84°25'16"~88°23'12"E), which is the southwestern part of the Tibetan Plateau. Mount Qomolangma is located on the border between China and Nepal. The blue rectangle area (Fig. 1) is the study area for the

glacial lake extraction in this study. The glaciers in the study area are cirque glaciers, which are distributed in depressions near the snow line (Ke et al. 2016). The annual precipitation in the area is less than 500 mm (Qi et al. 2013). Besides, no large rivers in the study area. The water supply of glacial lakes mainly relies on the melting water of ice and snow. Small streams developed by glacial lakes and glaciers in the study area are also marked in Fig. 1. The glacial lakes are mainly moraine-dammed lakes and

glacial erosion lakes (Cirque lakes), while the small glacial lakes are mainly moraine thaw lakes, accounting for the largest proportion in number.



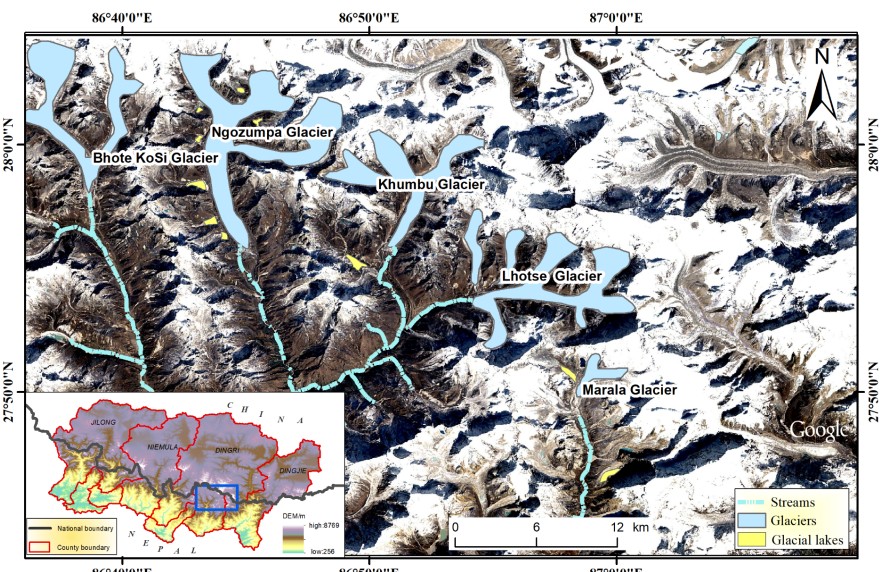

Figure 1. Location and topography information of the study area. All copyrights Image ©Google Earth 2020.

**2.2 Data**

**2.2.1 Dataset and preprocessing**

The global glacial lakes are mainly distributed in mountainous areas with many glaciers, including the Himalayas in Asia, the Buenos Aires Mountains in South America (Bourgois et al. 2016), the Alaska Mountains in North America (Rick et al. 2019), and the Alps in Europe (Huggel et al. 2002), etc. Among them, the Qomolangma has different kinds of glacial lakes, such as the glacial erosion lake and the moraine-dammed lake, including seven types of glacial lakes according to the classification system summarized by Yao et al. (2017) (Fig. 3). However, since the Qomolangma is located on the Tibetan Plateau, the glacial lakes developed in this area are plateau glacial lakes with complex topographical backgrounds, less vegetation, easy freezing, etc. Moreover, due to differences in climate, topography, and geological activities, glacial lake development areas on other continents differ in terms of ground background, distribution density, etc. For instance, the ground background of glacial lakes in the Alps has higher vegetation coverage. Since training datasets significantly influence the final result, collecting more samples of different types is of great help to enhance the stability and universality of the model (He et al. 2021). For the sake of increasing the diversity of the training dataset, except for the high Asia region, this study also collected some glacial lake samples from other continents.



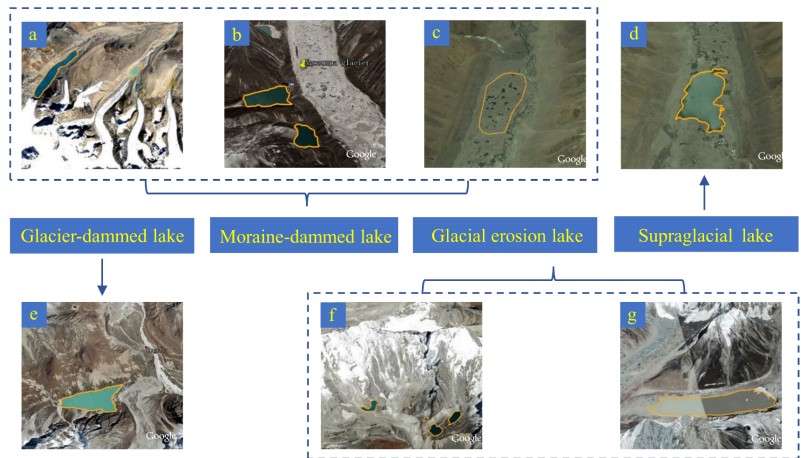

**Figure 2. Seven types of glacial lakes in the training dataset. All copyrights Image ©Google Earth 2020.**
Note: a) Terminal moraine-dammed lake, b) Side moraine-dammed lake, c) Moraine thaw lake, d) Supraglacial lake,
e) Glacial-dammed lake, f) Cirque lake, g) Glacier valley lake.

In the data preprocessing, 14 to 19 levels of Google Earth images were chosen for the glacial lake dataset, and the image resolution covers the range of 5 meters to sub-meters. When labeling, the glacial lakes were manually outlined with the help of ENVI 5.3, and every pixel in the image was labeled as 1 for glacial lakes or 0 for the background. When training the deep learning model, the images that were input into the network needed to be processed into image tiles for the limitation of the computer's memory capacity. After many experiments, it was more appropriate to divide the input images into non-overlapping image tiles of size 256×256. Image tiles that do not contain glacial lakes were removed to alleviate the problem of large background areas. Data augmentation operations were carried out to increase the number of samples, like image rotation. Finally, a total of 15376 training samples with a size of 256×256 were obtained, out of which 20% of image tiles were selected as validation data randomly (Table 1).

**Table 1. Details of the glacial lake training dataset based on Google Earth images in this study. All copyrights Image ©Google Earth 2020.**

| Continents | Area | Data level | Number of samples (256×256) | Sample examples (256×256) |
|---|---|---|---|---|
| Asia | Himalaya, Northern Tibet Plateau | Level14-18 (0.28-4.45m) | 5494 | |
| South America | Buenos Aires Mountains | Level 17、18 (0.4m, 0.79m) | 3397 | |
| North America | Alaska Mountains | Level 18、19 (0.14m, 0.28m) | 5519 | |
| Europe | the Alps | Level 18、19 (0.21m, 0.41m) | 966 | |



### 2.2.2 Other datasets

Except for the training dataset, other data products were also used to assist in completing this study
(Table 2). The second glacier inventory dataset of China was used to delineate the distribution area of
140 glacial lakes. In section 4.3, the 30 m glacial lake inventory in western China based on Landsat data, as
well as three global land-cover products (Gong et al. 2019) based on Sentinel images, were used for
comparison with the glacial lakes extracted in this study.

**Table 2. Other datasets were used in this study.**

| Dataset | Temporal coverage | Res. /m | Source |
|---|---|---|---|
| The second glacier inventory dataset of China (V1.0) | 2006-2011 | - | National Tibetan Plateau/Third Pole Environment Data Center (TPDC) http://www.tpdc.ac.cn/en/ |
| Inventory data of glacial lake in west China | 2015 | 30 | National Tibetan Plateau/Third Pole Environment Data Center (TPDC) http://www.tpdc.ac.cn/en/ |
| FROM-GLC10 | 2017 | 10 | Tsinghua University (THU) http://data.ess.tsinghua.edu.cn/ |
| ESA World Cover | 2020 | 10 | European Space Agency (ESA) https://viewer.esa-worldcover.org/worldcover |
| Esri Land Cover | 2020 | 10 | Environmental Systems Research Institute (ESRI) https://livingatlas.arcgis.com/landcover/ |

### 3 Methods

145 In the glacial lake extraction method, based on Google Earth images, this study used the semantic
segmentation framework to achieve rough extraction of the glacial lake first (output1 in Fig. 3). Then
two-level optimization combined Simple Linear Iterative Clustering (SLIC) and Dense Conditional
Random Field (DenseCRF) was used to achieve refined extraction of glacial lake outlines (output 2 and
output 3 in Fig. 3). By the way, these two optimization methods can also be used separately to implement
150 single-level optimization.



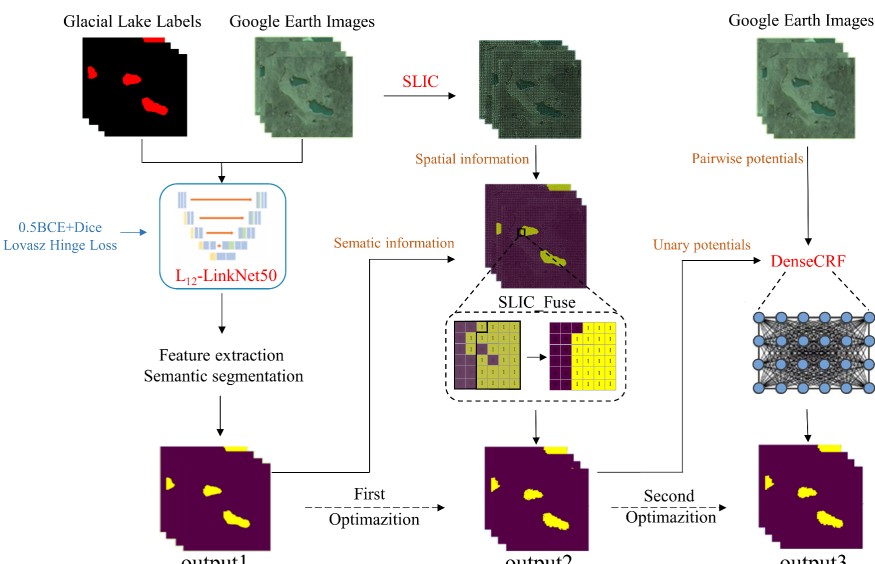

**Figure 3. Structure diagram of glacial lake extraction strategy in this study. All copyrights Image ©Google Earth 2020.**

## 3.1 Rough extraction of glacial lake information based on the semantic segmentation network

### 3.1.1 Deep Residual Network-LinkNet50

The LinkNet network (Chaurasia and Culurciello, 2017) uses ResNet18 (He et al. 2016) as the backbone of the U-Net (Sathananthavathi and Indumathi, 2021), which has a lightweight structure and fast calculation speed. Good results have also been achieved in identifying glacial lakes, but the effect is unsatisfactory in areas covered with ice and snow or with a small number of glacial lakes. Moreover, high-resolution images in the dataset built in this study have complex spatial and spectral information and deeper networks are more beneficial for high-level feature extraction (Li et al. 2020; Wang et al. 2021). Besides, LinkNet50 has achieved good results in road detection based on high-resolution images (Li and Liu, 2022). Therefore, to obtain more useful features to distinguish glacial lakes from the background, this study used a deep Residual Network (ResNet50) instead of ResNet18 as the backbone of U-Net.

As shown in Table 3, ResNet of different depths contains five stages, and the output results (feature images) of the second to fifth stages are Res2, Res3, Res4, and Res5. In ResNet50, their sizes (width × height ×channel) are $64 \times 64 \times 256$, $32 \times 32 \times 512$, $16 \times 16 \times 1024$, $8 \times 8 \times 2048$, respectively. On the right side in Fig. 4, ResNet50 is used in the Encoder of LinkNet50 for feature extraction to obtain high-level features. The input of each Encoder layer is also bypassed to the output of the corresponding Decoder layer. On the right side, the Decoder uses the residual structure to combine low-level and high-level features and recover the detailed information of the image lost by the down-sampling.





**Table 3. The structure of ResNet of different network depths.**

| Stage | Output feature map size (width × height) | ResNet18 | ResNet34 | ResNet50 | ResNet101 | ResNet152 |
|---|---|---|---|---|---|---|
| $C_1$ | 128×128 | A convolutional layer with the kernel of 7×7, the stride of 2 A max-pooling layer with the kernel of 3×3, the stride of 2 | | | | |
| $C_2$ | 64×64 | $\begin{bmatrix}3\times3,64\\3\times3,64\end{bmatrix}\times2$ | $\begin{bmatrix}3\times3,64\\3\times3,64\end{bmatrix}\times3$ | $\begin{bmatrix}1\times1,64\\3\times3,64\\1\times1,256\end{bmatrix}\times3$ | $\begin{bmatrix}1\times1,64\\3\times3,64\\1\times1,256\end{bmatrix}\times3$ | $\begin{bmatrix}1\times1,64\\3\times3,64\\1\times1,256\end{bmatrix}\times3$ |
| $C_3$ | 32×32 | $\begin{bmatrix}3\times3,128\\3\times3,128\end{bmatrix}\times2$ | $\begin{bmatrix}3\times3,128\\3\times3,128\end{bmatrix}\times4$ | $\begin{bmatrix}1\times1,128\\3\times3,128\\1\times1,512\end{bmatrix}\times4$ | $\begin{bmatrix}1\times1,128\\3\times3,128\\1\times1,512\end{bmatrix}\times4$ | $\begin{bmatrix}1\times1,128\\3\times3,128\\1\times1,512\end{bmatrix}\times8$ |
| $C_4$ | 16×16 | $\begin{bmatrix}3\times3,256\\3\times3,256\end{bmatrix}\times2$ | $\begin{bmatrix}3\times3,256\\3\times3,256\end{bmatrix}\times6$ | $\begin{bmatrix}1\times1,256\\3\times3,256\\1\times1,1024\end{bmatrix}\times6$ | $\begin{bmatrix}1\times1,256\\3\times3,256\\1\times1,1024\end{bmatrix}\times23$ | $\begin{bmatrix}1\times1,256\\3\times3,256\\1\times1,1024\end{bmatrix}\times36$ |
| $C_5$ | 8×8 | $\begin{bmatrix}3\times3,512\\3\times3,512\end{bmatrix}\times2$ | $\begin{bmatrix}3\times3,512\\3\times3,512\end{bmatrix}\times3$ | $\begin{bmatrix}1\times1,512\\3\times3,512\\1\times1,2048\end{bmatrix}\times3$ | $\begin{bmatrix}1\times1,512\\3\times3,512\\1\times1,2048\end{bmatrix}\times3$ | $\begin{bmatrix}1\times1,512\\3\times3,512\\1\times1,2048\end{bmatrix}\times3$ |
| | 1×1 | Average pooling layer, 1000-dimensional fully connected layer, the softmax function | | | | |

Note: The size of the input image is 256×256×3. In the matrix multiplication expressions (5 columns on the right), take ResNet18 as an example, where 3×3 indicates the convolution kernel size, 64 indicates the number of channels of the output image, and 2 indicates two residual blocks.

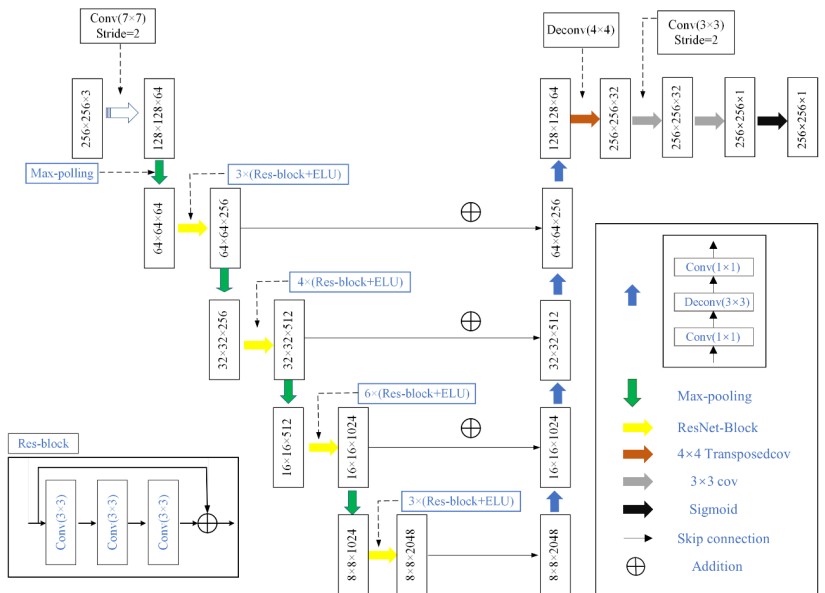

**Figure 4. Schematic diagram of the LinkNet50 network structure used in this study.**

**3.1.2 Loss function**

In the dataset in this study, the glacial lake area is small and the background area is large (unbalanced samples), so the target's features cannot be fully learned during the model training process. Therefore, to solve this problem, Dice Loss, which can help to reduce the impact of unbalanced positive and negative





samples in binary classification, was used in this study. Dice Loss essentially measures the overlap of two samples and the calculation formula is:

$$d = 1 - 2|X \cap Y|/(|X| + |Y|) \tag{1}$$

where $|X|$ and $|Y|$ indicates the number of pixels of sample $X$ and sample $Y$, respectively; $|X \cap Y|$ indicates the intersection of $X$ and $Y$. For the common part is repeatedly calculated, the coefficient of the $|X \cap Y|$ is 2.

However, Dice Loss will affect backpropagation, making the loss change unstable during model training. To increase the stability of the training process, BCE Loss was introduced in this study. BCE Loss belongs to the cross-entropy loss function, which is used to evaluate the difference between the probability distribution obtained by the training model and the natural distribution. In binary classification, the model predicted the probability of each category as p and 1-p, respectively, and the

loss function is:

$$L = \frac{1}{N}\sum_i L_i = \frac{1}{N}\sum_i -[y_i.\log(p_i) + (1 - y_i).\log(1 - p_i)] \tag{2}$$

where $y_i$ is the label of sample i, 1 for positive and 0 for negative. $p_i$ is the probability that sample i is predicted to be positive. After testing, we finally adopted $Loss1 = 0.5BCE + Dice$ as the loss function of the training model, which not only solved the problem of unbalanced positive and negative samples but also increased the stability of the network training process.

In addition, Lovasz Hinge Loss is a convex Lovasz extension of Submodular Losses, which could optimize the IoU loss of the network in the condition of unbalanced sample distribution (Berman et al. 2018). It is worth noting that the LinkNet50 with the Loss1 as the loss function is called L₁-LinkNet50 in this study. After L₁-LinkNet50, Lovasz Hinge Loss (Loss2) was further used to fine-tune the deep semantic segmentation network in this study, which is referred to L₁₂-LinkNet50.

**3.2 High-precision edge optimization algorithm**

**3.2.1 Simple linear iterative clustering (SLIC)**

SLIC is a superpixel segmentation algorithm proposed by Achanta et al. (2012) with the advantages of a simple calculation process, high computation speed, and good edge matching. First, it converts the image from RGB to CIE-Lab, in which the 5-dimensional vector V[l, a, b, x, y] consists of $(l, \ a, \ b)$ color value

and $(x, \ y)$ coordinates of the corresponding pixel. Then based on the idea of K-means, k superpixels are initialized in an image, and the distance between them is set as S. The core part is to iteratively calculate the centers of these superpixels by a clustering method. The distance for 5-dimensional vectors $(D')$ includes the distance of the Lab color space $(d_c)$ and the geometric space $(d_s)$. The following formulas are used to calculate the distance:

$$d_c = \sqrt{(l_j - l_i)^2 + (a_j - a_i)^2 + (b_j - b_i)^2} \tag{3}$$

$$d_s = \sqrt{(x_j - x_i)^2 + (y_j - y_i)^2} \tag{4}$$

$$D' = \sqrt{(d_c/m)^2 + (d_s/s)^2} \tag{5}$$

where $m$ indicates the maximum possible distance in the Lab color space; $s$ indicates the maximum possible value in the geometric space.



For each superpixel center, the range of pixel searching is 2S × 2S. If the distance from a pixel to the superpixel center i is less than the distance from it to the superpixel center to which it previously belonged, then this pixel is assigned to the superpixel i. Iteratively optimizes according to this algorithm until the superpixel center of each pixel no longer changes. After iterating out the superpixel segmentation blocks of the image, the semantic segmentation results and SLIC segmentation are fused

based on a rule. First, count the number of pixels with different semantic segmentation labels in a superpixel. Then the semantic label with more pixels is labeled to all pixels of this superpixel segmentation block.

### 3.2.2 DenseCRF

DenceCRF overcomes the limitation that CRF can only be performed in a local area and cannot connect

full-text information (Zhang et al. 2018). The global context information of the whole image is organically combined, and all the pixels in the entire image are connected with the current pixel. The DenseCRF is composed of unary potentials and pairwise potentials. Unary potentials come from the output of the front-end semantic segmentation network, which refers to the potential of predicting the pixel point(i) as a semantic label($x_i$) through the semantic segmentation network. Pairwise potentials

describe the relationship of each pixel to all other pixels in the image, mainly providing position and spectral information through the original input image (Berman et al. 2018). Therefore, it not only makes predictions for a single pixel but also calculates the probability of different classes appearing simultaneously.

### 3.3 Accuracy Assessment Indicators

Scientific selection of evaluation indicators is the key to testing the accuracy of glacial lake extraction results. Four indicators, Recall, Precision, F1 Score (Yacouby and Axman, 2020), and IoU (Rahman and Wang, 2016), were selected as the indicators for the accuracy evaluation of glacial lake extraction results. And all of them are generated based on the True Positive (TP), True Negative (TN), False Positive (FP), and False Negative (FN).

**Table 4. Meanings and calculation formulas of four evaluation indicators.**

| Evaluating Indicators | Formulas | Meanings |
|---|---|---|
| *Recall* | $TP/(TP + FN)$ | The proportion of the number of correctly identified samples to the number of all positive samples. |
| *Precision* | $TP/(TP + FP)$ | The proportion of the number of correctly identified samples to the number of predicted positive samples. |
| *F1 Score* | $2 * Precision * Recall/(Precisio + Recall)$ | The harmonic average of recall rate and accuracy rate. |
| *IoU* | $TP/(TP + FN + FP)$ | The ratio of intersection and union of real value and predicted value |



## 4 Experiments and Results

### 4.1 Comparative analysis of rough extraction results

Based on high-resolution Google Earth images (Level 18, with an image resolution of 0.52 m) in the study area, we used the $L_1$-LinkNet50 for the preliminary extraction in the rough extraction stage. The

commonly used semantic segmentation models (U-Net, LinkNet) and traditional machine learning methods (SVM and RF) were chosen for comparison. Moreover, among the improved semantic segmentation methods for glacial lakes, EfficientNet U-Net (Qayyum et al. 2020) was proposed based on high-resolution images (3-4 m), which was also chosen for comparison. Our experiment was based on Python3.6 and the open-source deep learning framework PyTorch. The training was performed on an

NVIDIA Geforce RTX 2080 Ti, using cuDNN10.0 for acceleration. The batch size was set to 4. The optimization method adopted the Adaptive Moment Estimation (Adam) and the initial learning rate was set to 1e-4. Moreover, the learning rate update strategy of polynomial decay was adopted to prevent the network from sinking into local optimal solutions later in model training, in which the momentum and weight decay were set at 0.9 and 1e-4, respectively. This study trained all networks for 45 epochs in this

stage.

Recall, Precision, F1 Score, and IoU were calculated to evaluate the extraction results against the ground truth obtained by manual digitization. As can be seen in Fig. 5, the edges of the glacial lakes extracted by SVM and RF are rough. There are difficulties in the complete extraction of complex glacial lakes and small glacial lakes, which affects the Recall. In deep learning models, the U-Net network is

greatly affected by snow, and the probability of being wrongly classified as glacial lakes is high, which decreases the Precision and F1 Score. EfficientNet U-Net obtains the highest Precision and the predicted water masks have few false positives, which is consistent with the conclusions of Qayyum et al. (2020). This method reduced false detections of glacial lakes in snow-covered areas compared to U-Net. However, there are still problems for glacial lakes with similar spectral information to the background,

and for glacial lakes that are shaded by mountain shadows. LinkNet can identify more glacial lakes than EfficientNet U-Net. But some false detections are prone to occur in the shaded area, and the Precision is reduced. Finally, after introducing the deep residual network, $L_1$-LinkNet50 improved the extraction of glacial lakes with small areas and glacial lakes shaded by mountain shadows. Although the Precision is slightly lower than that of EfficientNet U-Net, the final F1 Score of $L_1$-LinkNet50 reaches 87.77% and

the Recall is 3.46% higher than that of EfficientNet U-Net. Therefore, it can be found that $L_1$-LinkNet50 has the most vital comprehensive ability for glacial lake extraction in these models.

**Table 5. Quantitative evaluation for glacial lake extraction.**

| Method | Recall | Precision | F1 Score | IoU |
|---|---|---|---|---|
| SVM | 73.61% | 88.28% | 80.28% | 67.05% |
| RF | 74.38% | 89.32% | 81.17% | 68.31% |
| UNet | 79.88% | 80.50% | 80.19% | 70.33% |
| EfficientNet U-Net | 81.04% | **92.08%** | 85.72% | 77.70% |
| LinkNet | 83.65% | 88.44% | 85.97% | 75.02% |
| $L_1$-LinkNet50 | **84.50%** | 91.31% | **87.77%** | **78.21%** |



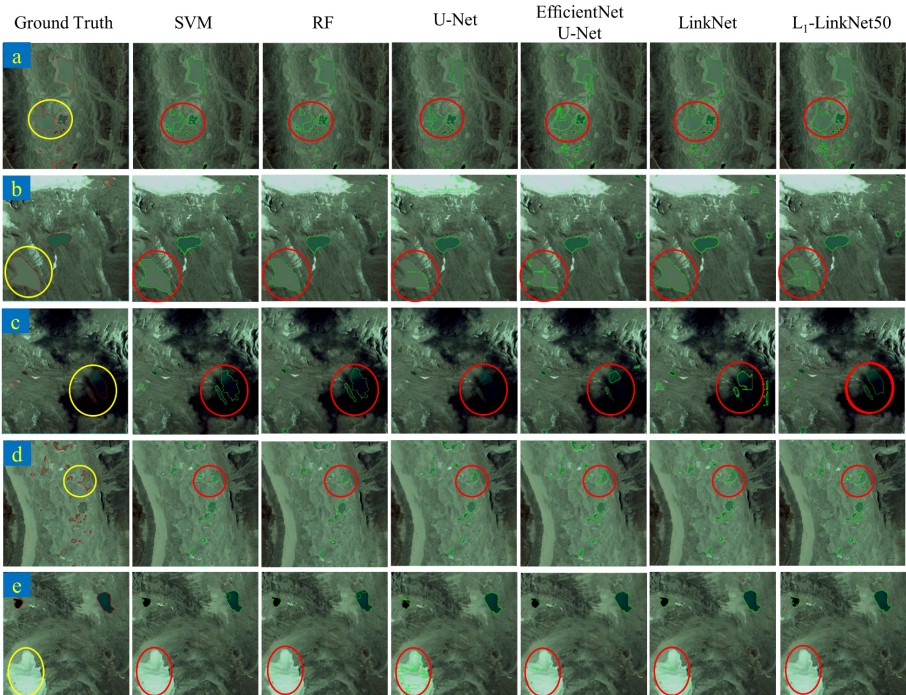

**Figure 5. Performance comparison of different models for the glacial lake extraction. All copyrights Image
©Google Earth 2020.**
Note: Five regions of the same size (2048×2048) were chosen based on Google Earth images (0.52 m), including
areas with glacial lakes of complex outlines (a), the inconsistent color of water bodies (b), mountain shadows (c),
and areas with multiple small glacial lakes (d), ice and snow (e).

**4.2 Comparative analysis of optimization results**

$L_1$-LinkNet50 showed the best effect to extract glacial lakes in section 4.1. This study improved $L_1$-
LinkNet50 and carried out post-processing on the glacial lake segmentation results. The $L_1$-LinkNet50
was trained for another 25 epochs for $L_{12}$-LinkNet50, using the Lovasz Hinge Loss (Loss2) as the loss
function. Then the two-level optimization strategy (SLIC and DenseCRF) was used to optimize semantic
segmentation results. For superpixel segmentation, the number of superpixels, compactness, and iteration
times were set as 2800, 60, and 10, respectively. Then the semantic segmentation result by $L_{12}$-LinkNet50
or the fusion result by SLIC and $L_{12}$-LinkNet50 was input into DenseCRF as the unary potential. In this
process, the Mean-Field approximation method was used for inference to minimize the potential function.

In Table 6, after $L_1$-LinkNet50 was trained with Lovasz Hinge Loss, the IoU reached 83.77% in the
study area, which was 5.56% higher than that of $L_1$-LinkNet50. It alleviated the problems of adhesions
(multiple glacial lakes nearby detected as one) (Fig. 6 - (3) and (4)) and missed detections (Fig. 6 - (2)).
In addition, it is difficult for the semantic segmentation network to recover all the lost spatial information
when upsampling, resulting in imprecise segmentation edges. The glacial lakes after superpixel
segmentation optimization are closer to the natural boundary, especially the small glacial lakes. Post-
processed results by DenseCRF had smoother edges and the Precision increased by 0.86%. Moreover,




after using two-level optimization (SLIC-DenseCRF), missed detections of glacial lakes were effectively reduced. Compared to $L_{12}$-LinkNet50, the IoU and F1 Score increased by 6.92% and 3.96%, respectively. The comparison of results based on different optimization algorithms proves that the post-processing based on SLIC-DenseCRF for deep learning semantic segmentation results can improve the accuracy of glacial lake extraction.

**Table 6. Evaluation indicators for glacial lake identification results under different optimization conditions.**

| Method | Recall | Precision | F1 Score | IoU |
|---|---|---|---|---|
| $L_1$_LinkNet50 | 84.50% | 91.31% | 87.77% | 78.21% |
| $L_{12}$-LinkNet50 | 88.26% | 92.85% | 90.50% | 83.77% |
| $L_{12}$-LinkNet50-SLIC | 92.66% | 89.77% | 91.19% | 85.11% |
| $L_{12}$-LinkNet50-DenseCRF | 89.38% | 93.71% | 91.49% | 84.23% |
| $L_{12}$-LinkNet50-SLIC-DenseCRF | **96.52%** | **92.49%** | **94.46%** | **90.69%** |

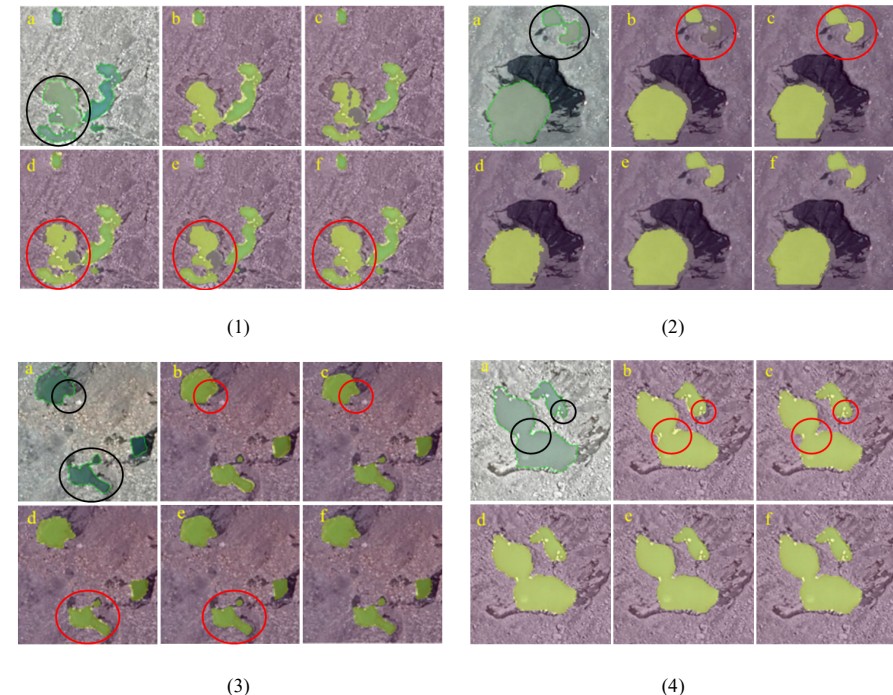

(1)         (2)

310   (3)         (4)

**Figure 6. Comparison of glacial lake identification results based on Google Earth images (0.52 m) under different optimization conditions. All copyrights Image ©Google Earth 2020.**
Note: Ground Truth (a), $L_1$-LinkNet50 (b), $L_{12}$-LinkNet50 (c), $L_{12}$-LinkNet50-SLIC (d), $L_{12}$-LinkNet50-DenseCRF (e), and $L_{12}$-LinkNet50-SLIC-DenseCRF (f).

**4.3 Glacial lake extraction in Qomolangma National Nature Reserve**

After evaluating the ability of the glacial lake extraction method proposed in this study, we applied $L_{12}$-LinkNet50-SLIC-DenseCRF to the extraction of glacial lakes in the Qomolangma National Nature Reserve (QNNR). This reserve has a total area of 33,819 km², including the core area, the buffer area,



and the experimental area. Because it is difficult to obtain the sub-meter level Google Earth image of the

entire QNNR, Google Earth images (level 16) in 2020 within the 10 km buffer zone from the end of the

glacier in QNNR were used as the data source, with an image resolution of 2.11 meters. For the glacial

lake extraction in QNNR, the evaluation results show that the Precision, F1 score, Recall, and IoU are

79.65%, 85.55%, 82.49%, and 70.20%, respectively. The IoU of the actual application is lower than the

rectangular study area, but the precision has reached more than 85%. The final result is shown in Fig. 7.

In the QNNR, glacier lakes are mainly distributed in the altitude range of 4000 m to 6000 m. The area

and number of glacial lakes are approximately normally distributed, and both peak at 5000-5500 m,

which is consistent with the research of Yang et al. (2019) and Zhang et al. (2021). The area and number

of glacial lakes at the peak account for 66.58% and 39.70% of all glacial lakes, respectively.

      Compared with the existing glacial lake inventory and three land cover datasets in section 2.2.2, the

glacial lakes extracted based on the proposed method are closer to the real in terms of the number and

area (Fig. 8 (a)). The area of the largest glacial lake (5.943 km$^2$) of the reserve extracted in this study is

consistent with the other four and is closest to the ground truth. For glacial lakes with an area greater

than 0.01 km$^2$, the distribution of the number of glacial lakes is similar for all datasets. However, for

small glacial lakes, due to the advantages of remote sensing image sources and methods, the accuracy of

the extraction results of glacial lakes in this study is significantly better than the other four existing

datasets. Moreover, we checked the results of the glacial lake extraction in the QNNR and found that the

smallest glacial lake that can be fully and correctly extracted has an area of about 160 m$^2$.

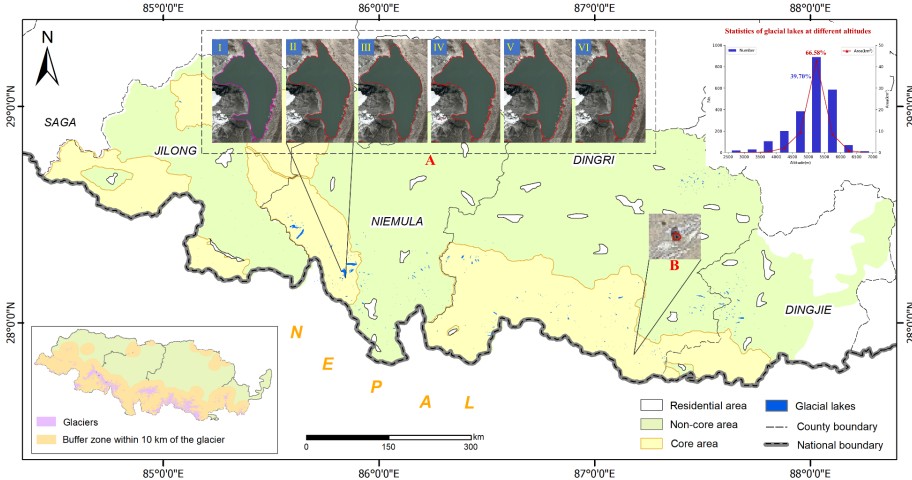

**Figure 7. Glacial lake extraction result in the QNNR based on Google Earth images in 2020 (2.11 m). All**
**copyrights Image ©Google Earth 2020.**
Note: A and B are the largest and smallest glacial lakes extracted in this region, respectively. Ground Truth (I), L12-
LinkNet50-SLIC-DenseCRF 2.11m (II), Inventory data of glacial lake 30m (III), ESA World Cover 10m (IV), Esri
Land Cover 10m (V), FROM-GLC10 10m (VI).




| No. | Number | Area/km² | Max/km² | Mini/m² |
|---|---|---|---|---|
| I | 1664 | 69.570 | 5.922 | 80 |
| **II** | **2300** | **65.170** | **5.943** | **160** |
| III | 599 | 64.279 | 5.409 | 3688 |
| IV | 23663 | 90.257 | 5.947 | 1886 |
| bV | 3032 | 85.016 | 6.190 | 1720 |
| VI | 11466 | 117.223 | 6.649 | 3135 |

(a)

| Area/km² | I | II | III | IV | V | VI |
|---|---|---|---|---|---|---|
| 1.0000-10.000 | 10 | 10 | 11 | 12 | 11 | 16 |
| 0.1000-1.0000 | 93 | 87 | 82 | 109 | 109 | 163 |
| 0.0500-0.1000 | 60 | 56 | 64 | 72 | 90 | 99 |
| 0.0300-0.0500 | 66 | 65 | 70 | 83 | 113 | 119 |
| 0.0100-0.0300 | 226 | 248 | 246 | 341 | 359 | 281 |
| 0.0050-0.0100 | 195 | 203 | 126 | 415 | 335 | 1103 |
| 0.0010-0.0050 | 533 | 634 | 0 | 2244 | 841 | 4034 |
| 0.0005-0.0010 | 232 | 345 | 0 | 2164 | 283 | 1618 |
| 0.0000-0.0005 | 249 | 652 | 0 | 18223 | 891 | 4033 |

(b)

**Figure 8. Comparison of glacial lakes between this study and the other four datasets.**

Note: Basic information (a), Statistical information on the number distribution of glacial lakes in different areas (b). Ground Truth (I), $L_{12}$-LinkNet50-SLIC-DenseCRF 2.11m (II), Inventory data of glacial lake 30m (III), ESA World Cover 10m (IV), Esri Land Cover 10m (V), FROM-GLC10 10m (VI).

## 5 Discussion

The lack of reliable glacial lake samples is one of the difficulties in the development of glacial lake extraction research based on deep learning networks (Wang et al, 2022). Qayyum et al. (2020) also stated that because of insufficient types of glacial lakes in the training dataset, some muddy brown glacial lakes could not be identified. The high resolution samples built in this study help to improve the evaluation indicators. Moreover, $L_{12}$-LinkNet50 uses a deep residual structure (ResNet50) as the backbone of the network, which enhances the ability to extract complex features, and is also better at the boundary of the small glacial lake (Fig. 9 B). Therefore, compared with other methods as shown in section 4.1, the evaluation indicators of glacial lakes extracted by $L_{12}$-LinkNet50 are improved. The proposed method in this study can effectively reduce missed detections of some glacial lakes that show similar spectral features with soil (Fig. 9 C) and shadows.

For post-processing, the parameter values used in the SLIC algorithm in this study, including the number of superpixel blocks (2800) and the compactness (60), were obtained through multiple experiments by the single-variable method based on sub-meter-level images. When the image resolution differs greatly, the amount of semantic information in a single superpixel will change. Thus, these parameter values are not applicable to images with a spatial resolution of ten meters such as Sentinel images. In addition, frozen lakes generally start from the edge with shallow water bodies and more small rocks, resulting in more noise on the edge of glacial lakes. The DenseCRF connects the local and global information to set up pair-wise potentials on all pairs of pixels, providing more detailed labeling and reducing the small-area noise generated by the image segmentation of high-resolution images. As a result, the optimized glacial lakes have smoother edges (Fig. 9 A) and fewer false spots on the lake surface.

For the glacial lake extraction results of the QNNR, the curve in Fig. 9 shows that the area distribution of small area glacial lakes is consistent with the results of manual digitization. However, although the proposed method is effective in identifying glacial lakes with similar spectral information to shadows, it is prone to misjudgment in small areas of shadows(Fig. 9 E). Because the area of these shadows is too small, little spectral and texture information on the background can be extracted, so it is

difficult to be distinguished by the method in this study. The number distribution of big-scale glacial

lakes is consistent with the results of manual digitization, but more large-area glacial lakes have not been

fully identified. The spectral information of glacial lakes completely covered by snow and some glacial

lakes that have been frozen for a long time is very similar to that of snow. At present, the proposed

method in this study cannot fully identify those glacial lakes (Fig. 9 D). Moreover, most of the Google

Earth Images in the QNNR were collected during the winter (for a small number of clouds) with large

snow-covered areas. This is also the reason why glacial lake evaluation indicators of the QNNR are lower

than those of the small study area in section 4.2.

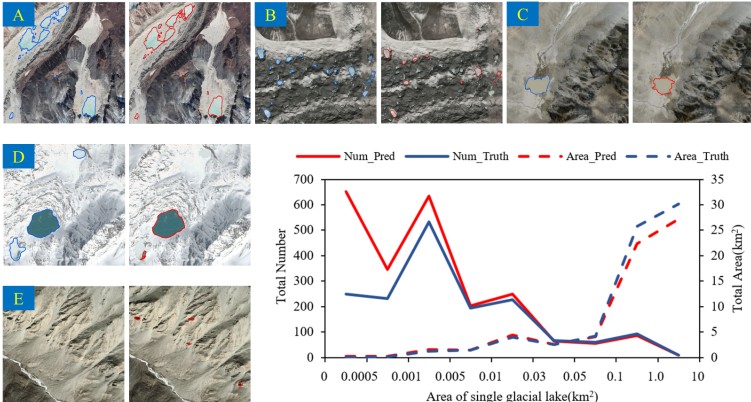

**Figure 9. The display of glacial lake extraction results in different situations in the QNNR, as well as the statistics of area and number. All copyrights Image ©Google Earth 2020.**

**6 Conclusions**

Aiming at the demand for high-accuracy outline extraction of glacial lakes in the high Asia region, this

study built a dataset for glacial lakes based on the global meter level to sub-meter level Google Earth

images; then proposed the glacial lake extraction method of $L_{12}$-LinkNet50 semantic segmentation

network with two-level optimization of SLIC-DenseCRF.

Based on the dataset containing glacial lakes of multiple types, the ability to identify glacial lakes

of different types and colors is improved in this study. 0.5BCE+Dice and Lovasz Hinge Loss are

combined to improve the loss function of the deep semantic segmentation network and suppress the

impact of the unbalanced positive and negative samples in the dataset. It has the advantage of small

glacial lake detection and effectively reduces the missed detection of glacial lakes that have similar

spectral features to the bare soil or shadows. The F1 Score in the study area reaches more than 90%. By

post-processing for the semantic segmentation results, the edges of glacial lakes are more consistent with

the actual situation, and the noise spots on the lake surface are also reduced.

Although the proposed method has achieved good extraction results on the new dataset, there are

still shortcomings in the recognition of snow covered glacial lakes and terrain shadows with small areas.

For future research, multi-source remote sensing images can be used to reduce the impact of snow cover

and shadows.



**Data availability**

All raw data can be provided by the corresponding authors upon request.

**Author contribution**

YC and MP designed the experiments;XB,RL, and PD prepared experimental data; XB and MP developed the code; YC and XB performed the data analysis. YC, XB, and MP reviewed and edited the manuscript.

**Competing interests**

The authors declare that they have no conflict of interest.

**Acknowledgments**

This research was funded by the National Natural Science Foundation of China [41771451] and the Sichuan Province Youth Science and Technology Innovation Team under Grant [2020JDTD0003]. Thanks to Gong P. et al., ESA, and ESRI for providing the land cover products (FROM- GLC10, ESA World Cover, and Esri Land Cover). And thanks to the TPDC for the second glacier inventory dataset of

China and the inventory data of the glacial lake in west China.

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
