# Peer review of "Refined glacial lake extraction in a high Asia region by Deep Neural Network and Superpixel-based Conditional Random Field"

_The Cryosphere, 2022_

## Referee Comment (RC1)

Review of 'Refined glacial lake extraction in high Asia by Deep Neural Network and Superpixel-based Conditional Random Field'

This manuscript proposed a refined methodology to extract glacial lakes from satellite imagery. Overall it is reasonably clear and well-written. Comments are below

Main comments

- Page 2 lines 39-44 - The discussion about semi-automatic methods is unclear. Please point out what makes these methods semi-automatic. For example, are these methods that use specified thresholds or feature as part of their workflow? Why are they more regionally restrictive than automatic methods. Please revise this part.

- Page 2 line 56 - Why is the method in Zhao et al. 2018 only applicable to Landsat images (is this due to the bands available, if so, please state this as it is more specific).

- Page 2 line 67 - 'with NDWI as the spatial attention' - what does it mean to have a feature as the attention mechanism? Or do you mean something different here? I could not find this in the paper (He et al. 2021). Did you mean to reference the paper 'J. Wang, F. Chen, M. Zhang and B. Yu, "NAU-Net: A New Deep Learning Framework in Glacial Lake Detection," in IEEE Geoscience and Remote Sensing Letters, vol. 19, pp. 1-5, 2022, Art no. 2000905, doi: 10.1109/LGRS.2022.3165045'?

- lines 124 -125 I usually use data, like Landsat, downloaded directly from e.g. NOAA or similar sources, and am not very familiar with Google Earth Engine. Other readers of this journal may be similar. I don't follow specifically what images are used or what the different levels (e.g. levels 14 to 19 of Google Earth images) mean - could you please elaborate?

- line 134 Could these image tiles randomly selected for validation be part of the same scene as those used for training data, and if so, how do you think this may impact your scores?

- lines 147-148 - For this audience I think more information is needed as to what SLIC and Dense CRF are, and why they are chosen.

- In Figure 3 - Are the arrows that indicate 'First optimization ' and 'Second optimization' steps (e.g., in a code) or is the first optimization the steps from labels/GEE images to output 1, and if so would it be better to put a box around this part and call it 'first optimization'. Similar comment applies to the 'second optimization'.

- lines 162-165 The reader might wonder what is so special about using LinkNet. It's a little hard to follow the motivation here - LinkNet is able to extract deep/complex features with fewer weights than a standard U-Net - is this essential for this problem? Or does it have to do with the addition property of LinkNet when the features from the encoder and decoder are combined?

- Figure 4 - tell the reader what the difference is between a deconv is in comparison to a transposed conv and why you have this distinction in your network.

- Both sections 3.2.1 and 3.2.2 don't give the reader an idea as to why a superpixel segmentation algorithm is needed, or why a conditional random field are chosen. This information is also not clear earlier (in introduction or background material). Hence it's hard to read these two sections and extract useful information. If these details aren't discussed further, the overview (i.e., why these two methods are chosen ) could be put earlier in the manuscript, while the details in 3.2.1 and 3.2.2 could be put in an appendix.

- lines 230 'full-text' information - can you place this in the context of the present study?

- No details about RF or SVM are given. We need more information to understand if this is a reasonable comparison. It these two methods were used in a previous study by the authors then they could refer to this here. These two could also be omitted since there is also a comparison to

UNet and EfficientNet UNet if adding that information would make the manuscript too long and the emphasis is not on feature learning vs feature engineering.

- lines 271-273 - Show in your figures where the false detections and shadows are. For example, figure 5 does not say what the red and yellow circles are referring to. A similar comment applies to the multi-color circles in Figure 6.

- I am not sure if contribution (2) on page 3 is considered a contribution since this loss function has been used in other studies. However, it may be the first time this has been used for segmentation of small objects? For example, a related (but different) approach was taken for the problem of sea ice floe segmentation ('Nagi, A.S.; Kumar, D.; Sola, D.; Scott, K.A. RUF: Effective Sea Ice Floe Segmentation Using End-to-End RES-UNET-CRF with Dual Loss. Remote Sens. 2021, 13, 2460. https://doi.org/10.3390/rs13132460') but without the Lovaz loss, which seems to make a significant different here. It might be worth pointing this out (perhaps in the conclusions) since the small lakes are somewhat similar to ice floes in that they are irregular small objects in background state.

- the conclusions refer to the high F1 score in 'the study area' but not (more generally) the results in the QNNR - why is this?

Minor comments

- In the abstract the authors refer to 'Google Earth' images - please be more specific about what images are used. In addition two different image resolutions (2.11 m and 0.52 m) are referred to but it's not sure why two different resolutions are used. They also refer to 'the study region' and then later (in the abstract) 'Qumolongma National Nature reserve', which adds to the confusion.

- abstract line 20 - if pixel spacing is 2.11 m then wouldn't $6 \times 6$ pixels be smaller than $160m^2$.

- page 2 line 1 'time and vigor' - A different word could be used than vigor - resources?

- page 2 line 62 'EfficintNet' typo

- page 2 line 63 'better result' - better than what?

- page 2 line 69 'area difference between positive and negative samples' - do you mean the difference between the area occupied by positive and negative samples (with there being more area for one than the other)?

- line 98 - remove 'Besides' before 'no large rivers', for example change to 'There are no large rivers in the study area'.

- It would be helpful to show the QNNR area in Figure 1 and then add text to the figure caption indicating which area is used for train/test and how the evaluation for the QNNR is done different that the entire study region (if I follow correctly only inference was done for the QNNR).

- The inset in Figure 7 is far too small.

- line 366 - I am not sure the reader will know what the 'single-variable' method is.

---

## Author Comment (AC1)

**Responses to AC1**

Dear referee,

We would like to express our sincere appreciation for your careful reading and helpful comments to improve the quality of this paper. Such comments are extremely valuable and helpful for revising and improving our paper. We apologize for the problems in our manuscript, and we have fixed them in the revised version. Our point-by-point responses to your comments are given as follows. The sentences in italic and underline are the comments.

**Main comments**

✧ **Comment 1:**

*Page 2 lines 39-44 - The discussion about semi-automatic methods is unclear. Please point out what makes these methods semi-automatic. For example, are these methods that use specified thresholds or feature as part of their workflow? Why are they more regionally restrictive than automatic methods. Please revise this part.*

**Respond:**

Thank you very much for your comment and valuable suggestion. The intraclass heterogeneity of glacial lakes is large, and glacial lakes in different regions tend to show greater differences. Therefore, when using traditional semi-automatic methods, it is necessary to manually adjust the index threshold or select more appropriate features according to the characteristics of glacial lakes and ground backgrounds in different regions. Besides, traditional machine learning relies on mathematical models for classification, and artificially set glacial lake labels and parameters have a great impact on the final results. For automatic methods such as deep learning, the computer automatically extracts features of the glacial lake and automatically updates parameters without human intervention. The following is the revised part.

Re-edit:

Due to the different elements contained in the water body of the glacial lake and the depth of the glacial lake, the glacial lake has a large intra-class heterogeneity, and different spectral information is displayed on the optical remote sensing image. Although these semi-automatic extraction methods are widely used, the operation is manually dependent and regionally restrictive, limiting the promotion and application on the global/hemispheric scale. For example, the glacial lakes in the Mount Everest area are mostly blue or green, while the glacial lakes developed in the deep valleys of the eastern section of Nyainqentanglha are closer to black. Therefore, when using traditional machine learning methods, it is necessary to artificially select appropriate features according to the characteristics of the glacial lake and ground background in the area.

✧ **Comment 2:**

*Page 2 line 56 - Why is the method in Zhao et al. 2018 only applicable to Landsat images (is this due to the bands available, if so, please state this as it is more specific).*

**Respond:**

The method uses the SWIR band of Landsat images, while most high-resolution images have only four bands of red, green, blue and near-infrared, and Google Earth images have only three bands. Therefore, it is written "only applicable to Landsat images". But after your kind and valuable reminder, we realized that the ten-meter-level Sentinel images also have the SWIR band, and it may be more appropriate to change it to "only applicable to remote sensing images with multiple bands (especially SWIR) such as Landsat images". The following is the revised part.
Re-edit:
It is only applicable to remote sensing images with multiple bands (especially SWIR) such as Landsat images, while most high-resolution images have only four bands of red, green, blue and near-infrared.

✧ **Comment 3:**

*Page 2 line 67 – 'with NDWI as the spatial attention' - what does it mean to have a feature as the attention mechanism? Or do you mean something different here? I could not find this in the paper (He et al. 2021). Did you mean to reference the paper 'J. Wang, F. Chen, M. Zhang and B. Yu, "NAU-Net: A New Deep Learning Framework in Glacial Lake Detection," in IEEE Geoscience and Remote Sensing Letters, vol. 19, pp. 1-5, 2022, Art no. 2000905, doi: 10.1109/LGRS.2022.3165045'?*

**Respond:**

For "with NDWI as the spatial attention", we mean to reference the paper "NAU-Net: A New Deep Learning Framework in Glacial Lake Detection". We are very sorry to cause you trouble here, we have modified the language expression here, and hope that it can explain clearly after the modification.
Re-edit:
He et al. (2021) added a space attention mechanism into the skip connection of U-Net to focus on glacial lakes. Wang et al. (2022b) proposed NAU-Net with NDWI as the spatial attention, which guided the network to pay more attention to the glacial lake information of low-level features and solved the problem of the area difference between the area occupied by positive and negative samples.

✧ **Comment 4:**

*lines 124 -125 I usually use data, like Landsat, downloaded directly from e.g. NOAA or similar sources, and am not very familiar with Google Earth Engine. Other readers of this journal may be similar. I don't follow specifically what images are used or what the different levels (e.g. levels 14 to 19 of Google Earth images) mean - could you please elaborate?*

**Respond:**

Thanks for your kind suggestion. Google Earth images is not a single data source, but a data collection of various aerospace imagery (Landsat, QuickBird, etc.) and aerial photography. Different levels of Google images correspond to different spatial resolutions, the higher the level, the higher the spatial resolution. For example, the spatial resolution of Google Earth imagery at level 19 is 0.14 meters, and at level 14 it is 4.45 meters. The following is a supplementary introduction to Google Earth imagery.

Re-edit:

==Google Earth imagery is a data collection of various aerospace and aerial remote sensing images, including Landsat , QuickBird, IKONOS images and other data. Different levels of Google Earth images have different spatial resolutions, and the higher the level, the higher the spatial resolution.==

✧ **Comment 5:**

*line 134 Could these image tiles randomly selected for validation be part of the same scene as those used for training data, and if so, how do you think this may impact your scores?*

**Respond:**

Thanks for your kind comment. In the glacial lake dataset, all image tiles from different continents are mixed together. The glacial lake samples were collected from glacial lakes in different scenes to ensure the richness and comprehensiveness of the glacial lake samples. Therefore, the training samples (80%) and validation samples (20%) randomly selected in the dataset during model training do not specifically correspond to a certain scene, but the overall training and validation results. We are very sorry to cause you trouble here, we have modified the language expression here, and hope that it can explain clearly after the modification.

Re-edit:

==Finally, a total of 15376 samples with a size of 256×256 were obtained, out of which 20% of image tiles were selected as validation data randomly (Table 1).==

✧ **Comment 6:**

*lines 147-148 - For this audience I think more information is needed as to what SLIC and Dense CRF are, and why they are chosen.*

**Respond:**

Thank you very much for your comment and valuable suggestion. The following is the revised part.

Re-edit:

==In the pixel-based semantic segmentation, the outline of the glacial lake is not refined enough, which does not fit the actual smooth edge of the glacial lake. The Simple Linear Iterative Clustering (SLIC) algorithm could fuse the rough result of semantic segmentation with the edge information of superpixel==

segmentation to enhance the integrity of the glacial lake and improve the edge segmentation. The Dense Conditional Random Field (DenseCRF) uses the constraint relationship between pixels to encourage similar pixels to be assigned the same label, while pixels with large differences are assigned different labels to obtain accurate glacial lake outlines. In this paper, after semantic segmentation, two-level optimization combined SLIC and DenseCRF was used to achieve refined extraction of glacial lake outlines (output 2 and output 3 in Fig. 3). By the way, these two optimization methods can also be used separately to implement single-level optimization.

✧ **Comment 7:**

*In Figure 3 - Are the arrows that indicate `First optimization ' and `Second optimization' steps (e.g., in a code) or is the first optimization the steps from labels/GEE images to output 1, and if so would it be better to put a box around this part and call it `first optimization'. Similar comment applies to the 'second optimization'.*

**Respond:**

The arrows in Figure 3 indicate "First optimization" and "Second optimization" steps. Based on your valuable suggestions, we added boxes to make the structure diagram more clear. Thanks again for your kind comment.
Re-edit:

[Figure]

**Figure 3. Structure diagram of glacial lake extraction strategy in this study. All copyrights Image ©Google Earth 2020.**

✧   **Comment 8:**

*lines 162-165 The reader might wonder what is so special about using LinkNet. It's a little hard to follow the motivation here - LinkNet is able to extract deep/complex features with fewer weights than a standard U-Net - is this essential for this problem? Or does it have to do with the addition property of LinkNet when the features from the encoder and decoder are combined?*

**Respond:**

Thank you for your suggestion, we have explained in the text corresponding position. After multiple down-sampling by Encoder, some spatial information is lost, and it is difficult to recover these lost spatial information in the Decoder part, which is not conducive to the extraction of glacial lakes. The commonly used U-Net network uses concat operation to combine the feature map obtained by the encoder part with the feature map corresponding to the decoder. However, the size of the feature map corresponding to the two layers is inconsistent, so the feature map obtained by the encoder needs to be cut to complete the splicing, and some effective information is lost. In the LinkNet network, the size of each layer feature map corresponding to Encoder and Decoder is the same, and the addition method is used to combine the features, and the shallow features are re-learned without increasing the parameters, so as to effectively restore the spatial information of the glacial lake.
Re-edit:
The LinkNet network (Chaurasia and Culurciello, 2017) uses ResNet18 (He et al. 2016) as the backbone of the U-Net (Sathananthavathi and Indumathi, 2021). In the LinkNet network, the size of each layer feature map corresponding to Encoder and Decoder is the same, and the addition method is used to combine the features, and the shallow features are re-learned without increasing the parameters, so that the spatial information of the glacial lake can be effectively restored, which has a lightweight structure and fast calculation speed.

✧   **Comment 9:**

*Figure 4 - tell the reader what the difference is between a deconv is in comparison to a transposed conv and why you have this distinction in your network.*

**Respond:**

Thanks for your kind comment. There is no difference in method between transposed convolution and deconvolution. In the pytorch code, both are nn.convtranspose2d(), only the parameters and uses are different. The following are revised notes to Figure 4.
Re-edit:
Note: In the Decoder, Conv(1×1) is responsible for reducing the number of channels(×1/4), and Deconv(3×3) only changes the size of the feature map (×2). After the decoder, Transposedconv (Deconv(4×4)) will reduce the number of channels (×1/2) and expand the size of the feature map(×2).

✧ **Comment 10:**

*Both sections 3.2.1 and 3.2.2 don't give the reader an idea as to why a superpixel segmentation algorithm is needed, or why a conditional random field are chosen. This information is also not clear earlier (in introduction or background material). Hence it's hard to read these two sections and extract useful information. If these details aren't discussed further, the overview (i.e., why these two methods are chosen ) could be put earlier in the manuscript, while the details in 3.2.1 and 3.2.2 could be put in an appendix.*

**Respond:**

Thank you very much for your comment and valuable suggestion. The reason for choosing a conditional random field has been updated in the revised part.
Re-edit:

In the pixel-based semantic segmentation, the outline of the glacial lake is not refined enough, which does not fit the actual smooth edge of the glacial lake. The Simple Linear Iterative Clustering (SLIC) algorithm could fuse the rough result of semantic segmentation with the edge information of superpixel segmentation to enhance the integrity of the glacial lake and improve the edge segmentation. The Dense Conditional Random Field (DenseCRF) uses the constraint relationship between pixels to encourage similar pixels to be assigned the same label, while pixels with large differences are assigned different labels to obtain accurate glacial lake outlines. In this paper, after semantic segmentation, two-level optimization combined SLIC and DenseCRF was used to achieve refined extraction of glacial lake outlines (output 2 and output 3 in Fig. 3). By the way, these two optimization methods can also be used separately to implement single-level optimization.

✧ **Comment 11:**

*lines 230 'full-text' information - can you place this in the context of the present study?*

**Respond:**

The ' full-text ' here means all the pixels in the image. The category of each pixel in the DenceCRF algorithm is related to the category of all other pixels in the image. DenceCRF connects all pairs of individual pixels in the image, enabling greatly refined segmentation and labeling. It means the same as the global context information in the following text.

✧ **Comment 12:**

*No details about RF or SVM are given. We need more information to understand if this is a reasonable comparison. It these two methods were used in a previous study by the authors then they could refer to this here. These two could also be omitted since there is also a comparison to UNet and EfficientNet UNet if adding that information would make the manuscript too long and the emphasis is not on feature learning vs feature engineering.*

**Respond:**

Thank you for your advice. We first use SLIC for superpixel segmentation, and perform feature extraction of pixel blocks to obtain training samples, and then use SVM and RF two traditional machine learning algorithms for classification. For the parameter settings of these two algorithms, we supplement them in the revised part. In the SVM classifier, the penalty coefficient ( C ) was set to be 12, the kernel to be radial basis function ( rbf ), and the gamma in kernel function to be 0.187. In the RF classifier, the number of decision trees was set to 150, and the number of features is set to 2.

Re-edit:

In the SVM classifier, the penalty coefficient ( C ) was set to be 12, the kernel to be radial basis function ( rbf ), and the gamma in kernel function to be 0.187. In the RF classifier, the number of decision trees was set to 150, and the number of features is set to 2.

✧ **Comment 13:**

*lines 271-273 - Show in your figures where the false detections and shadows are. For example, figure5 does not say what the red and yellow circles are referring to. A similar comment applies to the multi-color circles in Figure 6.*

**Respond:**

Thank you very much for your comment and valuable suggestion. The circle represents the difference between the results extracted by different methods and the real label ( yellow : real label, red : extraction result ), guide readers to focus here. In order to improve readability, we have unified the colors.

Re-edit:

[Figure]

**Figure 5. Performance comparison of different models for the glacial lake extraction. All copyrights Image ©Google Earth 2020.**

Note: Five regions of the same size (2048×2048) were chosen based on Google Earth images (0.52 m). The red vectors are the boundary of the truth glacial lakes, and the green vectors are the boundary of the extraction result, including areas with glacial lakes of complex outlines (a), the inconsistent color of water bodies (b), mountain shadows (c), and areas with multiple small glacial lakes (d), ice and snow (e).

[Figure]

(1)

(2)

(3)

(4)

**Figure 6. Comparison of glacial lake identification results based on Google Earth images (0.52 m) under different optimization conditions. All copyrights Image ©Google Earth 2020.**

Note: Ground Truth (a), $L_1$-LinkNet50 (b), $L_{12}$-LinkNet50 (c), $L_{12}$-LinkNet50-SLIC (d), $L_{12}$-LinkNet50-DenseCRF (e), and $L_{12}$-LinkNet50-SLIC-DenseCRF (f). ==The green vectors are the boundary of the truth glacial lakes, and the yellow masks are the semantic segmentation result.==

**✧ Comment 14:**

*I am not sure if contribution (2) on page 3 is considered a contribution since this loss function has been used in other studies. However, it may be the first time this has been used for segmentation of small objects? For example, a related (but different) approach was taken for the problem of sea ice floe segmentation ('Nagi, A.S.; Kumar, D.; Sola, D.; Scott, K.A. RUF: E_ective Sea Ice Floe Segmentation Using End-to-End RES-UNET-CRF with Dual Loss. Remote Sens. 2021, 13, 2460. https://doi.org/10.3390/rs13132460') but without the Lovaz loss, which seems to make a significant different here. It might be worth pointing this out (perhaps in the conclusions) since the small lakes are somewhat similar to ice floes in that they are irregular small objects in background state.*

**Respond:**

Thank you again for your positive comments and valuable suggestions. After using the loss function of 0.5BCE + Dice to train the network, we use Lovasz Hinge Loss to fine-tune the LinkNet50 network. Our difference lies in the use of a two-step constrained loss function and training strategy. Therefore, we modified the expression in the manuscript.

Re-edit:

(2) To alleviate the negative impact of unbalanced positive and negative samples on the network extraction for glacial lake features, ==a two-step constrained loss function and training strategy were proposed with Resnet50 as the backbone.==

**✧ Comment 15:**

*the conclusions refer to the high F1 score in 'the study area' but not (more generally) the results in the QNNR - why is this?*

**Respond:**

Thank you for your suggestion, we are not comprehensive summary of the article. We have made modifications to the corresponding part.

Re-edit:

The F1 Score in the study area reaches more than 90%. ==At the same time, it is applied to a wider range of QNNR. Due to the misjudgment of the small glacial lake in the shadow, the F1 score is reduced, but it also reaches 82.49 %.==

**Minor comments**

✦ **Comment 1:**

*In the abstract the authors refer to `Google Earth' images - please be more specific about what images are used. In addition two different image resolutions (2.11 m and 0.52 m) are referred to but it's not sure why two different resolutions are used. They also refer to `the study region' and then later (in the abstract) `Qumolongma National Nature reserve', which adds to the confusion. reply:*

**Respond:**

Thanks for your kind suggestion. Google Earth images is not a single data source, but a data collection of various aerospace imagery (Landsat, QuickBird, etc.) and aerial photography. Different levels of Google images correspond to different spatial resolutions, the higher the level, the higher the spatial resolution. 2.11 m and 0.52 m are the highest spatial resolutions of Google Earth images available in this area. The study region is used to verify the feasibility of the method. After proving that the method is feasible, the glacial lake is extracted from a larger area (Qumolongma National Nature Reserve ), which belongs to the application area.

✦ **Comment 2:**

*abstract line 20 - if pixel spacing is 2.11 m then wouldn't 6 ×6 pixels be smaller than 160m². *

**Respond:**

Thank you for your careful reading and apologize for our inappropriate expression. We have made changes in the corresponding position.
Re-edit:
The area of the minimum glacial lake that can be extracted is 160 m2 (less than 6×6 pixels).

✦ **Comment 3:**

*page 2 line 1 'time and vigor' - A different word could be used than vigor - resources?*

**Respond:**

We sincerely thank the reviewer for careful reading. As suggested by the reviewer, we have corrected the 'time and vigor' to 'time and resources'.
Re-edit:
Still, it costs lots of time and resources, which is challenging to meet the needs for large-scale glacial lake identification.

**❖ Comment 4:**

*page 2 line 62 'EfficintNet' typo*

**Respond:**

We are very sorry for our careless mistake and it was rectified.
Re-edit:
Qayyum et al. (2020) used the pre-trained ==EfficientNet== as the backbone of the U-Net to map glacial lakes,

**❖ Comment 5:**

*'better result' - better than what?*

**Respond:**

We are very sorry for our unclear expression. and it was rectified.
Re-edit:
Qayyum et al. (2020) used the pre-trained ==EfficientNet== as the backbone of the U-Net to map glacial lakes, which achieved a better result ==than the original U-Net, RF and SVM classifiers== in high-resolution glacial lakes extraction.

**❖ Comment 6**

*page 2 line 69 `area difference between positive and negative samples' - do you mean the difference between the area occupied by positive and negative samples (with there being more area for one than the other)?*

**Respond:**

Thank you for your careful reading and suggestions, and sorry that our expression is not clear, we have made changes.
Re-edit:
==Wang et al. (2022b) proposed NAU-Net with NDWI as the spatial attention, which guided the network to pay more attention to the glacial lake information of low-level features and solved the problem of the area difference between the area occupied by positive and negative samples.==

**❖ Comment 7**

*line 98 - remove 'Besides' before 'no large rivers', for example change to `There are no large rivers in the study area'.*

**Respond:**

We think this is an excellent suggestion and we have made changes to this.

Re-edit:

There is no large rivers in the study area.

✧ **Comment 8**

*It would be helpful to show the QNNR area in Figure 1 and then add text to the figure caption indicating which area is used for train/test and how the evaluation for the QNNR is done different that the entire study region (if I follow correctly only inference was done for the QNNR).*

**Respond:**

Thank you for your advice. The data set used for training and testing in the paper contains samples from multiple regions in Asia, South America, North America and Europe. After the model training is completed, the glacial lake is predicted in the area of Figure 1. The analysis shows that the L12-LinkNet50-SLIC-DenseCRF model has the best effect, and then the model is applied to QNNR to test whether it can be applied to a large area. If the entire dataset location is displayed, a global-scale base map needs to be drawn, so that our study area is difficult to be found in the map. At the same time, since the QNNR boundary coincides with most of the county-level administrative boundaries, we did not add the QNNR vector to the figure. We annotated the same county name in Figure 7 ( main picture ) and Figure 1 ( lower left ), and the two images can be connected through the county.

✧ **Comment 9**

*The inset in Figure 7 is far too small.*

**Respond:**

Thank you very much for your correction and patience. We have noticed this problem and modified it at the corresponding location.

Re-edit:

[Figure]

**Figure 7. Glacial lake extraction result in the QNNR based on Google Earth images in 2020 (2.11 m). All copyrights Image ©Google Earth 2020.**

Note: A and B are the largest and smallest glacial lakes extracted in this region, respectively. The purple vectors are the boundary of the truth glacial lakes, and the red vectors are the boundary of the extraction result. Ground Truth (I), $L_{12}$-LinkNet50-SLIC-DenseCRF 2.11m (II), Inventory data of glacial lake 30m (III), ESA World Cover 10m (IV), Esri Land Cover 10m (V), FROM-GLC10 10m (VI).

✧ **Comment 10**

*line 366 - I am not sure the reader will know what the 'single-variable' method is.*

**Respond:**

Thank you for your advice. What we want to express is that the two parameters superpixel blocks and the compactness that SLIC needs to adjust are tested separately. We have supplemented the manuscript. This means that when the compactness remains constant but the number of superpixel blocks varies, the optimal number of superpixel segmentations is selected. Similarly, the setting of compactness follows the same approach. We replace it with the control variable method.
Re-edit:
were obtained through multiple experiments by the control variable method based on sub-meter-level images.

---

## Author Comment (AC2)

**Responses to AC2**

Dear referee,

We would like to express our sincere appreciation for your careful reading and helpful comments to improve the quality of this paper. Such comments are extremely valuable and helpful for revising and improving our paper. We apologize for the problems in our manuscript, and we have fixed them in the revised version. Our point-by-point responses to your comments are given as follows. The sentences in italic and underline are the comments.

**Main comments**

✧ **Comment 1:**

*The introduction as a whole lacks clarity into why glacial lakes are important in the global context and the technical detail of different delineation methods are very complex and are difficult to follow for those new to the subject matter. See minor comments to try and help mitigate these problems.*

- On the whole, the first paragraph about the importance of glacial lakes is very limited and only makes statements with no explanation at all (i.e. why do glacial lakes have a strong relationship with climate change?).
- Outburst floods are not necessarily caused by 'melting glaciers' so I would strongly suggest this is changed.
- I really struggled to follow the second paragraph in the introduction (L47 to L58). It currently reads as if there is an assumption that the reader has prior knowledge about these techniques. More context is required to explain these technical approaches. The minor comments may help to resolve this problem.
- Why does a new paragraph begin on L47 for the automated approaches, when the authors present the manual and semi-automated approaches in the previous where they were originally mentioned?

**Respond:**

Thank you for your suggestion. For the introduction, we explain the relationship between glacial lakes and climate change. The cause of the glacial lake outburst is that we do not express it clearly enough, which will cause the reader that the melting of glaciers is a necessary condition for the glacial lake outburst. According to your suggestion, we have made corresponding adjustments to

this part of the statement.

Because we want to focus on the method of automatic extraction, including the deep learning method used in this paper, it also belongs to automatic extraction, so we will start another section of automatic extraction method. At the same time, this can also avoid the problem of putting the three methods in a paragraph that causes the current paragraph to be too long.

Re-edit:

Glacial lakes are natural water bodies mainly supplied by glacier meltwater or formed by water accumulation in moraine ridge depressions and are densely distributed in high Asia (Yao et al. 2017). Glacial lakes have a strong relationship to ongoing climate warming (Pandey et al. 2021). On the one hand, in the global warming environment, glaciers show a trend of melting, the glacier runoff increases, and the melting water into glacial lakes will lead to an increase in the area of surrounding glacial lakes or the occurrence of glacial lake outburst floods (GLOF). This may threaten the lives and property of surrounding residents (Song et al. 2016; Begam et al. 2018). The GLOF in Tibet on June 26,2020 caused 43.9 kilometers of roads, 8 bridges to be washed away, and 19.98 hectares of farmland to be flooded (Zheng et al. 2021). On the other hand, as a product of global warming and glacier melting, glacial lakes are one of the sensitive indicators reflecting global changes (Lei et al. 2014; Qiu et al. 2019; Zhou et al. 2019). Additionally, many glacial lakes are small in size and unevenly distributed (Yang et al. 2018). Small glacial lakes are more active and sensitive to climate change (Sakai et al. 2015; Zhang et al. 2015). Therefore, accurate monitoring of glacial lakes is essential to studies on global climate change, water resource distribution, and disaster warnings.

For the glacial lake outline delineation, there are mainly manual digitization methods, semi-automatic methods, and automatic methods. First, the manual digitization method has achieved good results (Yang et al. 2019). Still, it costs lots of time and resources, which is challenging to meet the needs for large-scale glacial lake identification. For the semi-automatic extraction of glacial lakes, current studies are mainly based on water body indices (Li et al. 2020) and machine learning(Jain et al. 2015; Veh et al. 2018). In 2015, Jain et al. (2015) used the Support Vector Machine (SVM) to detect glacial lakes in Bhutan, Himalayas. In 2018, Veh et al. (2018) trained a Random Forest (RF) classifier based on Landsat data and detected glacial lake outbursts through change detection technology. Due to the different elements contained in the water body of the glacial lake and the depth of the glacial lake, the glacial lake has a large intra-class heterogeneity, and different spectral information is displayed on the optical remote sensing image. Although these semi-automatic extraction methods are widely used, the operation is manually dependent and regionally restrictive, limiting the promotion and application on the global/hemispheric scale. For example, the glacial lakes in the Mount Everest area are mostly blue or green, while the glacial lakes developed in the deep valleys of the eastern section of Nyainqentanglha are closer to black. Therefore, when using traditional machine learning methods, it is necessary to artificially select appropriate features according to the characteristics of the glacial lake and ground background in the area.

✧ **Comment 2:**

*I would consider having 2.0 as 'Study area' as a stand-alone section and moving '2.2. Data' into the Methods section and re-naming '3.0 Methods and Data'. It currently seems strange to have the data within the study area section.*

**Respond:**

Thank you very much for your suggestion. According to your description, you propose to take the ' study area ' as an independent part and put the ' data ' into the method part. However, we believe that the current placement of data in the study area is more in line with the logic and process of the article and can provide readers with clearer reading guidance. Since data is the basis and core of the study, we believe that placing it in the study area can highlight the background and basic information of the study. And, putting the data in the method part may lead to the part becoming very long. Therefore, we believe that the existing structure is appropriate for explaining the background and data information of the research.

✧ **Comment 3:**

*In the data section, can the 'levels' of Google Earth images be explained? I don't tend to use these types of data when using openly accessible optical imagery and I'm sure other readers will not have used them either. What are they? Are they like bands on Landsat collections? Please clarify.*

**Respond:**

Thanks for your kind suggestion. Google Earth images is not a single data source, but a data collection of various aerospace imagery (Landsat, QuickBird, etc.) and aerial photography, so it doesn 't have a fixed temporal resolution. Different levels of Google images correspond to different spatial resolutions, the higher the level, the higher the spatial resolution. For example, the spatial resolution of Google Earth imagery at level 19 is 0.14 meters, and at level 14 it is 4.45 meters. The following is a supplementary introduction to Google Earth imagery.

Re-edit:
Google Earth imagery is a data collection of various aerospace and aerial remote sensing images, including Landsat , QuickBird, IKONOS images and other data. Different levels of Google Earth images have different spatial resolutions, and the higher the level, the higher the spatial resolution.

✧ **Comment 4:**

*Can ENVI 5.3 please be clarified to what it is and how it assisted the authors in the labelling process. Again, I don't use this software and I'm sure others haven't, so it would be good to note what it is and how it assisted labelling as this part is vital to the training of the dataset.*

**Respond:**

ENVI(The Environment for Visualizing Images) is not a software specifically designed for label making. It is a powerful remote sensing image processing software developed by scientists in the field of remote sensing using the interactive data language IDL ( Interactive Data Language ). It is a leading software solution to extract information from images quickly, conveniently and accurately. Today, many image analysts and scientists choose ENVI to extract information from remote sensing

images. ENVI has been widely used in scientific research, environmental protection, meteorology, petroleum and mineral exploration, agriculture, forestry, medicine, national defense & security, earth science, public facilities management, remote sensing engineering, water conservancy, ocean, surveying and mapping, urban and regional planning and other fields. 5.3 is the version number of the software.

We use ENVI software to draw data labels by manually sketching ROI(region of interesting ). Of course, other geographic information software can also achieve this function.

Re-edit:

When labeling, the glacial lakes were manually outlined with the help of ENVI (other software, such as LabelMe and ArcGIS, can also do labeling), and every pixel in the image was labeled as 1 for glacial lakes or 0 for the background.

✧ **Comment 5**

*It is important to note the spatial and temporal resolution (could be done in L145) of the Google Earth images so the reader understands explicitly. How often are images acquired? Is it sparse, or is it continuous? What times of the year is the analysis being undertaken (i.e. July to September)? What years have available data (i.e. 2010, 2015)?*

**Respond:**

Google Earth images is not a single data source, but a data collection of various aerospace imagery (Landsat, QuickBird, etc.) and aerial photography, so it doesn 't have a fixed temporal resolution. At the same time, this may also lead to differences in the minimum resolution available in different regions.

✧ **Comment 6**

*Is L248 to 260 (first results paragraph) not just methods? The reader is being told what version of Python is being used and the specific package?*

**Respond:**

Thank you for your suggestions. This paragraph gives an overview of the data, methods and operating environment of the experiments we used. In order not to bring misunderstanding to the reader, we move this passage to the top of Sect. 4.1 Comparative analysis of rough extraction results as an overview text.

✧ **Comment 7**

*All maps need more detail, north arrow, labels, scale bar. Also, figure captions are very limited (i.e. why are there different colour circles in Fig 5 and Fig 6?).*

**Respond:**

Thank you for your attention to the details and issues of the map in this article. For the details of the map, we will further increase the north arrow, label, scale and other information according to your recommendations to improve the readability and accuracy of the map. Figure 5 and Figure 6 are images used to explain the effect of automatic extraction, not normative maps, so we do not use north arrow, label and scale. And, we will unify the circle colors in Figure 5 and Figure 6 in the revision.

Re-edit:

[Figure]

**Figure 5. Performance comparison of different models for the glacial lake extraction. All copyrights Image ©Google Earth 2020.**

Note: Five regions of the same size (2048×2048) were chosen based on Google Earth images (0.52 m). The red vectors are the boundary of the truth glacial lakes, and the green vectors are the boundary of the extraction result, including areas with glacial lakes of complex outlines (a), the inconsistent color of water bodies (b), mountain shadows (c), and areas with multiple small glacial lakes (d), ice and snow (e).

[Figure]

**Figure 6. Comparison of glacial lake identification results based on Google Earth images (0.52 m) under different optimization conditions. All copyrights Image ©Google Earth 2020.**

Note: Ground Truth (a), $L_1$-LinkNet50 (b), $L_{12}$-LinkNet50 (c), $L_{12}$-LinkNet50-SLIC (d), $L_{12}$-LinkNet50-DenseCRF (e), and $L_{12}$-LinkNet50-SLIC-DenseCRF (f). The green vectors are the boundary of the truth glacial lakes, and the yellow masks are the semantic segmentation result.

✧ **Comment 8**

*It took me a long time to try and understand what figure 8(b) is actually showing and I'm still not convinced I do. What is this heat map actually showing? There is no label on the colour bar gradient?*

**Respond:**

I am very sorry to trouble you. We have noticed your question about Figure 8 ( b ) and have modified the original image. In the revised draft, labels will be added to the color bar to clarify the meaning expressed by the heat map. We apologize for taking a lot of time to understand this diagram.

Re-edit:

[Figure]

**Figure 8. Comparison of glacial lakes between this study and the other four datasets.** (b)

✧ **Comment 9**

*The discussion is quite limited in terms of the importance of the study and why this method is worthwhile.*

    o I'd suggest there needs to be some expansion on why we care about this approach. For example, the paper clearly states small lakes can be identified, so does this mean they are the smallest lakes ever to be identified? If so, how much total lake volume/area have we been missing from global inventories? Could a comparison be made to these global inventories?

**Respond:**

Thank you for your advice. In Figure 8 of Section 4.3 of the manuscript, we give the minimum glacial lake area that can be identified based on this method and compare it with the current four products. In the discussion, we focus on the analysis of the problems existing in the method of this manuscript. Because our method also has the situation of missed detection and false reduction, it cannot give a specific missed detection area or quantity compared with the existing catalog data. This is also a direction for future work. However, we can determine that the minimum glacial lake area for completely correct identification is increased from a few thousand km2 to 160 km2.

✧ **Comment 10**

*On a glaciological note, I think it would be worth having a figure or table with the seven lake types (as shown in Figure 2) and how each model performs for each category? It would help the discussion, particularly where it is noted on L395.*

**Respond:**

Thank you for your advice and feel sorry. Although we have collected different types of glacial lake images, this is only to enrich our samples. However, when we draw the sample set, we do not subdivide the glacial lakes, but only divide the images into two types of glacial lakes ( 1 ) and background ( 0 ), so the experimental results can not be classified more carefully. If the sample is redrawn, it takes a lot of time. At the same time, this may be the direction we can improve in the future.

✧ **Comment 11**

*Could I ask the authors to go and check abbreviations and ensure they have had their full title spelt out prior being abbreviated.*

**Respond:**

Thank you very much for your suggestion. We attach great importance to your attention to the use of abbreviations. In the revised version, we will carefully examine the use of abbreviations again and ensure that their full names are given before abbreviations. If any omissions or errors are found, we will modify them in time. We apologize for the inconvenience. Thank you for your criticism and guidance, we will strive to improve and provide more accurate and complete content.

✧ **Comment 12**

*Consider use of 'besides' throughout manuscript.*

**Respond:**

Thank you very much for your suggestions. We will carefully examine and correct the abbreviations used in the revision. In addition, we will also consider using the word ' besides ' throughout the manuscript. Thank you for your attention and valuable comments on our research.

**Minor comments**

✧ **Comment 1**

*In the title, should there be an 'a' between 'in' and 'high'?*

**Respond:**

Thank you for your advice. We have made changes in the corresponding position according to your opinion.

Re-edit:

Refined glacial lake extraction in a high Asia region by Deep Neural Network and Superpixel-based Conditional Random Field

✧ **Comment 2**

*L10: Would use the term to the 'glacial lake outburst flood' – depends on where glaciers/lakes are situated in proximity to towns/infrastructure to be considered 'disasters'*

**Respond:**

Thank you for your suggestion, here is our wording is not accurate. After discussion, we change disaster into event.

Re-edit:

Remote sensing extraction of glacial lakes is an effective way of monitoring water body distribution and outburst events.

✧ **Comment 3**

*L16: What is IoU – be careful with abbreviations*

**Respond:**

Thank you very much for your suggestions. We will carefully examine and correct the abbreviations used in the revision.

Re-edit:

In addition, Lovasz Hinge Loss is a convex Lovasz extension of Submodular Losses, which could optimize the IoU(Intersection Over Union) loss of the network in the condition of unbalanced sample distribution (Berman et al. 2018).

✧ **Comment 4**

*Remove two examples of 'about' on L18 and L20. Be specific.*

**Respond:**

Thank you for your advice. Because in the process of area calculation, in order to make the data more concise, we retain the last two decimal places for the data through rounding rules, so the word 'about ' is used here. We have deleted 'about' at your suggestion.

Re-edit:

With the Google Earth images of 2.11 m resolution in the Qomolangma National Nature Reserve, 2300 glacial lakes with a total area of 65.17 km$^2$ were detected by the proposed method. The area of the minimum glacial lake that can be extracted is 160 m$^2$ (less than 6×6 pixels).

◇ **Comment 5**

*L30: Why? Why are glacial lakes becoming larger as a result of increased warming? Statement needs to be explained.*

**Respond:**

In the global warming environment, glaciers show a trend of melting, the glacier runoff increases, and the melting water into glacial lakes will lead to an increase in the area of surrounding glacial lakes or the occurrence of glacial outburst floods.

◇ **Comment 6**

*L31: Besides does not correctly link to previous sentence*

**Respond:**

Thank you for your suggestion. We changed the connectives and revised the manuscript.

Re-edit:
Additionally, many glacial lakes are small in size and unevenly distributed (Yang et al. 2018).

◇ **Comment 7**

*L32: Why? Why are small lakes 'more active' and 'sensitive to climate change'? The last couple of 'why' comments are important as this is the glaciological rationale for studying these lakes. From the introduction so far, I'm questioning why glacial lakes are important as only statements have been made with no explanation.*

**Respond:**

Under the background of global warming, Zhang et al.(2019) analyzed the results of the Third Pole region glacial lake changes from 1990 to 2010, and concluded that the smaller lakes exhibited more significant area changes. Lake sizes of 0.05-0.2 km2 also exhibited more rapid area changes, i.e., > 6 %, over the last decade.

◇ **Comment 8**

*L34: I appreciate the need for real-time monitoring of glacial lakes and their risk downstream particularly. However, the introduction so far has not cited any examples of glacial lake outburst floods and the subsequent impacts?*

**Respond:**

Thank you for your advice. In the manuscript editing stage, we focused on the extraction of the glacial lake boundary, and did not elaborate on the impact of the glacial lake outburst. According to your requirements, we describe the impact of glacial lake outburst in the corresponding part of the introduction.

Re-edit:

The GLOF in Tibet on June 26,2020 caused 43.9 kilometers of roads, 8 bridges to be washed away, and 19.98 hectares of farmland to be flooded (Zheng et al. 2021).

✧ **Comment 9**

*L35: I think for the first instance of using a different word to extraction for the reader. I understand what you mean, but it could raise questions of what is being extracted? Area, volume? Would suggest rephrasing to 'For the glacial lake outline delineation' to simplify the sentence.*

**Respond:**

Thank you for your suggestion. I apologize for the confusion caused by the use of the word .Your proposed rephrasing, "For the glacial lake outline delineation", is a simpler and clearer way to express the intended meaning. We will make the necessary changes to the sentence to address this issue. Thank you for helping improve the clarity of the text.

Re-edit:

For the glacial lake outline delineation, there are mainly manual digitization methods, semi-automatic methods, and automatic methods.

✧ **Comment 10**

*Citation needed after 'results' on L37.*

**Respond:**

Thank you very much for pointing out. We have modified and added the corresponding references.

Re-edit:

First, the manual digitization method has achieved good results (Yang et al. 2019).

✧ **Comment 11**

*Vigour? Just state that it is time expensive and is therefore not suitable for global lake identification*

**Respond:**

We sincerely thank the reviewer for careful reading. As suggested by the reviewer, we have corrected the 'time and vigor' to 'time and resources'.

Re-edit:

Still, it costs lots of ==time and resources==, which is challenging to meet the needs for large-scale glacial lake identification.

✧ **Comment 12**

*Citation needed after machine learning on L39.*

**Respond:**

Thank you very much for pointing out. We have modified and added the corresponding references.

Re-edit:

For the semi-automatic extraction of glacial lakes, current studies are mainly based on water body indices (Li et al. 2020) and machine learning==(Jain et al. 2015; Veh et al. 2018)==.

✧ **Comment 13**

*L49: Remove canny*

**Respond:**

Thank you for your suggestions on my manuscript. I will follow your advice and delete the words you specified in the manuscript. At the same time, I will ensure that the fluency and accuracy of the manuscript is not affected.

Re-edit:

==The edge detection algorithm is one of the most classical and advanced image edge detection algorithms (Chen, 2021)==.

✧ **Comment 14**

*L51 to L52: This sentence makes no sense. What lake extraction is being referred to, is it from a study? What threshold? How is it automatically defined? Why did it not successfully identify lakes? Change sentence as it is not currently followable*

**Respond:**

Thank you for your advice. The purpose of these two sentences is to introduce the Threshold and Simplified C-V ( TSCV ) method. The threshold method corresponds to the water index method in

the semi-automatic method above. In order to avoid repetition and bring trouble to readers, we have deleted this sentence.\

✧ **Comment 15**

_L53: Again, what threshold? If it is going to be referred too, it needs to be stated at the top of the paragraph so the reader understands what this threshold is - i.e. is it a band threshold?_

**Respond:**

Thank you for reading carefully. Threshold method is a commonly used remote sensing image processing method for segmentation and extraction of ground objects and backgrounds in images. This method is based on the gray value of the image or the reflectivity of a specific band, and distinguishes the ground object and the background area by setting a threshold. For optical images, the water index is often calculated and a threshold is set to distinguish the water from the background. If the water index image is regarded as a band, it also belongs to the band threshold.

✧ **Comment 16**

_L55: why is it complicated?_

**Respond:**

Thank you for your question. The C-V level set method is a process of iterative evolution to continuously find the minimum value of the energy functional, so its convergence speed is very slow. It takes a long time to evolve to obtain the final image segmentation result, and the calculation amount is very large. There are similar expressions in the references(Zhao et al. 2018).

✧ **Comment 17**

_L56: Why is it limited to Landsat? Is it limited to all collections? Similar to entire introduction so far, it's too vague of a statement and needs to be more explicit._

**Respond:**

The method uses the SWIR band of Landsat images, while most high-resolution images have only four bands of red, green, blue and near-infrared, and Google Earth images have only three bands. Therefore, it is written "only applicable to Landsat images". But after your kind and valuable reminder, we realized that the ten-meter-level Sentinel images also have the SWIR band, and it may be more appropriate to change it to "only applicable to remote sensing images with multiple bands (especially SWIR) such as Landsat images". The following is the revised part

Re-edit:

It is only applicable to remote sensing images with multiple bands (especially SWIR) such as Landsat images, while most high-resolution images have only four bands of red, green, blue and near-infrared.

✧ **Comment 18**

*L63: Is there any numbers to show it was 'better'? Vague currently*

**Respond:**

In the experiment of Qayyum et al. (2020), show that the U-Net with EfficientNet backbone achieved the highest F1 Score of 0.936 than the U-Net with VGG style layers, RF and SVM classifiers.

✧ **Comment 19**

*L64: Sentence makes no sense – is this a follow-on point from the previous?*

**Respond:**

Thank you for your suggestion. This has indeed been mentioned above(line 32). In order to avoid repetition, we have deleted this sentence according to your requirements.

Re-edit:
Based on PlanetScope Imagery, Qayyum et al. (2020) used the pre-trained EfficientNet as the backbone of the U-Net to map glacial lakes, which achieved a better result than the original U-Net, RF and SVM classifiers in high-resolution glacial lakes extraction. Skip connection structure will transfer a large amount of redundant background information from the low level to the high level, reducing the utilization efficiency of low level features.

✧ **Comment 20**

*L65: What is 'skip connection structure'?*

**Respond:**

Skip connection structure will transfer a large amount of redundant background information from the low level to the high level, reducing the utilization efficiency of low level features. We have explained what is skip connection structure in the manuscript, and give a more detailed explanation in the following citations. In order to avoid the manuscript being too long, we have only made a simple explanation here.

**✧ Comment 21**

*L66: What are 'low level features'?*

**Respond:**

Low-level features are minor details of the image, like edge or color, that can be pickup by a convolutional filter, SIFT and HOG (Girshick et al. 2014).

**✧ Comment 22**

*L66: NDWI needs stating before abbreviating*

**Respond:**

Thank you very much for your advice. We will carefully examine each word abbreviation in the manuscript and ensure that it is presented in the form of a full name when it first appears. This will enable readers to better understand and read the manuscript.
Re-edit:
Wang et al. (2022b) proposed NAU-Net with Normalized Difference Water Index (NDWI) as the spatial attention, which guided the network to pay more attention to the glacial lake information of low-level features and solved the problem of the area difference between the area occupied by positive and negative samples.

**✧ Comment 23**

*L67: What is a 'space attention mechanism'?*

**Respond:**

The essence of the attention mechanism is to locate the information of interest and suppress the useless information. Not all regions in the image contribute equally to the task, only the task-related regions need to be concerned, such as the main body of the classification task. The spatial attention model is to find the most important part of the network for processing. (He et al. 2021; Wang et al. 2022b).

**✧ Comment 24**

*L70: State spatial resolution of Google Earth images*

**Respond:**

Thank you for your suggestion, we have explained in the data section.

✧ **Comment 25**

*L75: 'et al'???*

**Respond:**

Thank you very much for your advice. We will examine it carefully and delete the word ' et al ' at the end of the example in the manuscript. This can make the examples more concise and clear, and more in line with the reader 's reading habits.
Re-edit:
Although the texture of the water body was complex, resulting in more noise in the segmentation, the study showed that the deeper network achieved better performance than U-Net, DeepLab V3+ (Li et al. 2019).

✧ **Comment 26**

*L76: end-to-end? Remember readers may not have any prior knowledge of machine learning*

**Respond:**

Thank you for your advice. End-to-end describes a process that takes a system or service from beginning to end and delivers a complete functional solution, usually without needing to obtain anything from a third party. The citation explains what is an end-to-end network. In order to avoid the manuscript being too long, it is not explained in detail here.

✧ **Comment 27**

*L92: Remove 'undertaken in'*

**Respond:**

Thank you for your suggestions on my manuscript. I will follow your advice and delete the words you specified in the manuscript. At the same time, I will ensure that the fluency and accuracy of the manuscript is not affected.

Re-edit:
The study area is the Mount Qomolangma area (27°08'09" N~29°19'14"N, 84°25'16"~88°23'12"E)

✧ **Comment 28**

*L97: So? Why does the presence of rivers matter?*

**Respond:**

Because there are no large rivers in the study area, the water in the glacial lakes usually comes from the melting glaciers or the water formed by snowfall and rainfall.

✧ **Comment 29**

*L105 to L119: remove all examples of 'etc'*

**Respond:**

Thank you for your suggestions on my manuscript. I will follow your advice and delete the words you specified in the manuscript. At the same time, I will ensure that the fluency and accuracy of the manuscript is not affected.

Re-edit:

The global glacial lakes are mainly distributed in mountainous areas with many glaciers, including the Himalayas in Asia, the Buenos Aires Mountains in South America (Bourgois et al. 2016), the Alaska Mountains in North America (Rick et al. 2019), and the Alps in Europe (Huggel et al. 2002). However, since the Qomolangma is located on the Tibetan Plateau, the glacial lakes developed in this area are plateau glacial lakes with complex topographical backgrounds, less vegetation, easy freezing. Moreover, due to differences in climate, topography, and geological activities, glacial lake development areas on other continents differ in terms of ground background, distribution density.

✧ **Comment 30**

*L110: refer to Fig 2 after 'seven types' so the reader knows what type of lakes are being categorized*

**Respond:**

Thank you for your suggestion, here is our mistake, caused the reference picture reference error. We have made corresponding modifications in the manuscript.

Re-edit:

Among them, the Qomolangma has different kinds of glacial lakes, such as the glacial erosion lake and the moraine-dammed lake, including seven types of glacial lakes according to the classification system summarized by Yao et al. (2017) (Fig. 2).

✧ **Comment 31**

*L118: Why not in high Asia?*

**Respond:**

In order to ensure the diversity of samples, we not only collected samples in high Asia, but also collected images of glacial lakes in other regions.

✧ **Comment 32**

*L124: First mention of 'levels' of Google Earth images as noted in major comments. Please address.*

**Respond:**

Thank you for your suggestion, we have explained in the data section.

✧ **Comment 33**

*L125: Sub-meters? Be specific and use numbers*

**Respond:**

Thank you very much for your suggestions . In your comments, you mentioned asking for a more specific description of ' Sub-meters ' and the use of specific numbers. I will describe ' Sub-meters ' more specifically and use specific numbers to support our statement. This allows readers to understand the relevant concepts and data more clearly.

Re-edit:

In the data preprocessing, 14 to 19 levels of Google Earth images were chosen for the glacial lake dataset, and the image resolution covers the range of 5 meters to sub-meters(0.14 meters).

✧ **Comment 34**

*L128: Inputted?*

**Respond:**

Thank you for pointing out the word spelling errors in the article. We have made corrections and will make corrections in the revision.

Re-edit:

When training the deep learning model, the images that were inputted into the network needed to be processed into image tiles for the limitation of the computer's memory capacity.

✧ **Comment 35**

_L130: Why are images divided into 256X256? Is it to lessen the computational expense on storage?_

**Respond:**

Firstly, most of the neural network models require that the input image size is fixed. Secondly if the image is directly input into the deep learning network, it will lead to memory overflow, so it is necessary to cut the image into image blocks and input them into the network. By trying many different sizes ( $256 \times 256, 512 \times 512, 1024 \times 1024$ ), we found that the network recognition effect of $256 \times 256$ image block training is better. Therefore, the training and prediction image is divided into $256 \times 256$ sub-image blocks in the experiment.

✧ **Comment 36**

_L133: Why did you assume 20% of the training data was suitable for validation? Does this coverage provide enough assurances that the model can produce realistic results? Was it for time or computational reasons?_

**Respond:**

Thank you for your question. The decision to allocate 20 % of the training data for validation is not based solely on time or computational reasons. On the contrary, it is a common practice in machine learning to strike a balance between having enough data for training and having a separate data set for verification.
The purpose of the validation set is to evaluate the performance of the model on invisible data and ensure that it can be well generalized to new instances. By using part of the training data for verification, we aim to obtain the performance estimation of the model on data that has never been encountered before.

✧ **Comment 37**

_L140: Be specific – which Landsat collections?_

**Respond:**

Thank you for your advice, we have detailed instructions in the corresponding location

Re-edit:
In section 4.3, the 30 m glacial lake inventory in western China based on Landsat TM/ETM+/OLI

**✧ Comment 38**

*L141: Be specific – which Sentinel collections (1,2,3)?*

**Respond:**

Thank you for your advice, we have detailed instructions in the corresponding location

Re-edit:

as well as three global land-cover products (Gong et al. 2019) based on Sentinel-2 images, were used for comparison with the glacial lakes extracted in this study.

**✧ Comment 39**

*L149: Remove phrases like 'by the way'*

**Respond:**

Thank you for your suggestion, The following is the revised part.

Re-edit:

These two optimization methods can also be used separately to implement single-level optimization.

**✧ Comment 40**

*159: Why is it 'unsatisfactory' in snow and ice areas? Surely this is important and needs explaining to failures in the model*

**Respond:**

Thank you for your comments. The original UNet network structure is suitable for processing medical images with a single background and low complexity, and it is difficult to solve the problem of difficult separation of glacial lakes and background. We concluded in the experiment that the glacial lake covered by ice and snow is not ideal to directly use the original UNet network to extract the results, so we improved the network structure for the above problems.

**✧ Comment 41**

*L248: Again, with levels. Move spatial resolution up the manuscript*

**Respond:**

Thank you for your suggestion, we have explained in the data section.

✧ **Comment 42**

*L263: Rough? What is meant by rough?*

**Respond:**

Thank you for your questions. What we want to express is that the glacial lake boundary extracted by SVM and RF methods is not smooth, so we use the word rough instead.

✧ **Comment 43**

*L263: What difficulties? Snow cover?*

**Respond:**

Thank you for your questions. It is difficult to extract complex glacial lakes and small glacial lakes, which results in a low recall value of this method.

✧ **Comment 44**

*Table 5: Why are some numbers bolded? Is it highest scores? Explain in caption*

**Respond:**

Regarding your question about the bolded numbers in Table 5, you are absolutely correct. The bolded numbers represent the highest scores in the table. We have now included a caption in the manuscript to clarify this.

Re-edit:

**Table 5. Quantitative evaluation for glacial lake extraction.**

| Method | Recall | Precision | F1 Score | IoU |
|---|---|---|---|---|
| SVM | 73.61% | 88.28% | 80.28% | 67.05% |
| RF | 74.38% | 89.32% | 81.17% | 68.31% |
| UNet | 79.88% | 80.50% | 80.19% | 70.33% |
| EfficientNet U-Net | 81.04% | **92.08%** | 85.72% | 77.70% |
| LinkNet | 83.65% | 88.44% | 85.97% | 75.02% |
| $L_1$-LinkNet50 | **84.50%** | 91.31% | **87.77%** | **78.21%** |

Note: The bold font represents the highest score

**✧ Comment 45**

*L319: Why is it difficult to obtain?*

**Respond:**

Because Google image is a collection of multiple images, it can obtain different minimum resolutions in different places, depending on the limitations of the image itself.

**✧ Comment 46**

*Figure 7: Cannot see bar chart inset*

**Respond:**

Thank you very much for your correction and patience. We have noticed this problem and modified it at the corresponding location.

Re-edit:

[Figure]

**Figure 7. Glacial lake extraction result in the QNNR based on Google Earth images in 2020 (2.11 m). All copyrights Image ©Google Earth 2020.**

Note: A and B are the largest and smallest glacial lakes extracted in this region, respectively. The purple vectors are the boundary of the truth glacial lakes, and the red vectors are the boundary of the extraction result. Ground Truth (I), $L_{12}$-LinkNet50-SLIC-DenseCRF 2.11m (II), Inventory data of glacial lake 30m (III), ESA World Cover 10m (IV), Esri Land Cover 10m (V), FROM-GLC10 10m (VI).

✧ **Comment 47**

*L355: Is this sentence stating that the training data did not contain enough variety of glacial lakes, and therefore brown lakes were unable to be identified by the model? If so, please rephrase*

**Respond:**

Because in previous studies, the training data did not contain enough types of glacial lakes, brown lakes could not be identified by the model. Therefore, when making training data, we incorporate such glacial lakes into training data in order to improve the extraction accuracy of glacial lakes.

✧ **Comment 48**

*L358: What are evaluation indicators?*

**Respond:**

The evaluation metrics include Precision, F1 score, Recall, and IoU (in Sect. 3.3).

✧ **Comment 49**

*L359: What do you consider complex?*

**Respond:**

Thank you for your question. The complexity here is that the heterogeneity between glacial lakes and other land types is small ( some glacial lakes are similar to the background ), and the extraction of such glacial lakes ( similar to the background ) is difficult ( complex extraction ). What we want to express is that the ability of glacial lake feature recognition is enhanced, and the ability to extract features is enhanced. Therefore, we modified it to the glacial lake extraction ability under different background conditions.

Re-edit:

Moreover, $L_{12}$-LinkNet50 uses a deep residual structure (ResNet50) as the backbone of the network, which enhances the ability to extract glacial lakes under different background conditions, and is also better at the boundary of the small glacial lake (Fig. 9 B).

✧ **Comment 50**

*L362: Quote the percentage of reduced misses*

**Respond:**

Thank you for your advice. Applying the data set produced in the manuscript and the proposed method to the QNNR region to extract the glacial lake boundary can alleviate the problem that the muddy brown glacial lake cannot be identified in the above research. However, since we only used the method proposed in the manuscript to extract the glacial lake in the QNNR region and did not compare it with other methods, the percentage of missed detection cannot be reduced for the time being.

✧ **Comment 51**

*L367: Differs by how much?*

**Respond:**

Thank you for your questions. The resolution of the image is different, and the information contained in a single pixel block may be different. However, how much the resolution difference will cause such a change depends on the size of the object in the remote sensing image. For example, a large lake, where the image resolution changes from 0.5 meters to 10 meters, the pixel may still be water information, but for the edge source area of the lake, most of the information presented by the pixel may change from water to land surface.

✧ **Comment 52**

*L368: Sentinel 1,2,3?*

**Respond:**

Thank you for your suggestion, we have supplemented the detailed information.

Re-edit:

Thus, these parameter values are not applicable to images with a spatial resolution of ten meters such as ==band 2,3,4 of Sentinel-2 images==.

✧ **Comment 53**

*L377: misjudge*

**Respond:**

Thank you for pointing out the word spelling errors in the article. I have made corrections and will make corrections in the revision.

Re-edit:
However, although the proposed method is effective in identifying glacial lakes with similar spectral information to shadows, it is prone to misjudge in small areas of shadows(Fig. 9 E).

✧ **Comment 54**

*L379: Large-scale instead of big?*

**Respond:**

Thank you for your advice, we have made the appropriate changes in accordance with your request.

Re-edit:

The number distribution of large-scale glacial lakes is consistent with the results of manual digitization, but more large-area glacial lakes have not been fully identified.

✧ **Comment 55**

*L384: First mention of temporal aspect in 'winter' – see prior comments about temporal resolution of data*

**Respond:**

Thanks for your kind suggestion. Google Earth images is not a single data source, but a data collection of various aerospace imagery (Landsat, QuickBird, etc.) and aerial photography, so it doesn 't have a fixed temporal resolution. Therefore, it is impossible to give an accurate data acquisition time for the entire QNNR region. From the surface state of the glacial lake, it is speculated that most of them are collected in the winter time range.

**References:**

Girshick R., Donahue J., Darrell T. and Malik J.: Rich Feature Hierarchies for Accurate Object Detection and Semantic Segmentation. IEEE Conference on Computer Vision and Pattern Recognition, 580-587. https://doi.org/10.1109/CVPR.2014.81, 2014.

He, Y., Yao, S., Yang, W., Yan, H. W., Zhang, L. F., Wen, Z. Q., and Liu, T.: An extraction method for glacial lakes based on Landsat-8 imagery using an improved U-Net network. IEEE J. Sel. Top. Appl. Earth Obs. Remote Sens., 14, 6544-6558. https://doi.org/10.1109/JSTARS.2021.3085397, 2021.

Jain, S. K., Sinha, R. K., Chaudhary, A., and Shukla, S.: Expansion of a glacial lake, Tsho Chubda, Chamkhar Chu Basin, Hindukush Himalaya, Bhutan. Can. Geotech. J., 75(2): 1451-1464. https://doi.org/10.1007/s11069-014-1377-z, 2015.

Qayyum, N., Ghuffar, S., Ahmad, H. M., Yousaf, A., and Shahid, I.: Glacial lakes mapping using

multi satellite PlanetScope imagery and deep learning. ISPRS Int. J. Geoinf., 9: 560. https://doi.org/10.3390/ijgi9100560, 2020.

Veh, G., Korup, O., Roessner, S., and Walz, A.: Detecting Himalayan glacial lake outburst floods from Landsat time series. Remote Sens. Environ., 207: 84-97. https://doi.org/10.1016/j.rse.2017.12.025, 2018.

Wang, J., Chen, F., Zhang, M., and Yu, B.: NAU-Net: A New Deep Learning Framework in Glacial Lake Detection. IEEE Geosci. Remote Sensing Lett., 19, 1-5. https://doi.org/10.1109/LGRS.2022.3165045, 2022b.

Yang, C. D., Wang, X., Wei, J. F., Liu, Q. H., Lu, A. X., Zhang, Y., and Tang, Z. G.: Chinese glacial lake inventory based on 3S technology method. Journal of Geographical Science, 74(3):544-556. https://doi.org/10.11821/dlxb201903011, 2019.

Zhao, H., Chen, F., and Zhang, M. M.: A systematic extraction approach for mapping glacial lakes in high mountain regions of Asia. IEEE J. Sel. Top. Appl. Earth Obs. Remote Sens., 11(8): 2788-2799. https://doi.org/10.1109/JSTARS.2018.2846551, 2018.

Zheng, G. X., Mergili, M., Emmer, A., Allen, S., Bao, A., Guo, H., and Stoffel, M.: The 2020 glacial lake outburst flood at Jinwuco, Tibet: causes, impacts, and implications for hazard and risk assessment, The Cryosphere, 15, 3159–3180, https://doi.org/10.5194/tc-15-3159-2021, 2021.

---

## Author Response (AR2)

**Responses to referee and editor**

**Dear referee and editor,**

**We sincerely thank you again for your careful reading and helpful comments that improved the quality of this paper. We apologize for the problems in our manuscript, and we have fixed them in the revised version. Our point-by-point responses to your comments are given as follows. The sentences in italic and underline are the comments.**

✧ **Comment 1:**

*There is an abbreviation in the abstract (IoU) which is not defined (L17). Please do.*

**Respond:**

Thank you for your suggestion. In the abstract, we use the abbreviation IoU (Intersection over Union) but do not give the full name. We will clarify this in the revised paper. Thanks again for your feedback. And table 4 provides a detailed explanation of IoU.

Re-edit:

1) With Google Earth images of 0.52 m resolution in the study area, the Recall, Precision, F1 Score, and Intersection Over Union (IoU) of glacial lake extraction based on the proposed method are 96.52%, 92.49%, 94.46%, and 90.69%, respectively.

✧ **Comment 2:**

*L27: Please can it be clarified in the text what the 'strong relationship' is between glacial lakes and ongoing climate warming? This is just a statement. Is there a relationship between warming and lake size, and frequency? Clarify in text.*

**Respond:**

Thank you for your suggestion. Climate warming, continuous glacier retreat and ablation of differences in the debris cover have led to the formation of a large number of glacial lakes and the continuous expansion of glacial lake areas. Climate change affects both size and number of glacial lakes. In the revised version, we added some literature to support this point of view.

Re-edit:

Climate warming, continuous glacier retreat and ablation of differences in the debris cover have led to the formation of a large number of glacial lakes and the continuous expansion of glacial lake areas (Nie et al. 2017, Chen et al. 2021a). In the past 30 years, the number of glacial lakes in High Asia has increased by 17.4%, the total area has increased by 17.3%, and the glacial lake area in the whole region expanded by 0.58%/a (Zhang et al.2022a). The rapid change of glacial lakes may increase the possibility of the occurrence of glacial lake outburst floods (GLOFs) (Zhong et al. 2021). The risk of GLOFs in High Asia is the highest (Taylor et al.2023). This may threaten the lives and property of 30 surrounding residents, and downstream infrastructures (Song et al. 2016; Begam et al. 2018; Nie et al. 2023). Such as, the GLOF in Tibet on June 26,2020, led to the destruction of 43.9 kilometers of roads and 8 bridges, and the flooding of 19.98 hectares of farmland (Zheng et al. 2021). Therefore, continuous dynamic monitoring of glacial lakes is essential to studies on climate change, water resource distribution, and disaster warnings. However, many small and unevenly distributed glacial lakes are ignored, these glacial lakes usually have a high risk of outburst (Zhang et al. 2022b).

✧ **Comment 3:**

*Don't think 'on the other hand' is needed on L27 or L34 - the following statements are not contradicting.*

**Respond:**

Thanks again for your advice. We will re-examine the wording in these two places and make appropriate adjustments in the revised paper. Thanks again for your feedback.

Re-edit:

Climate warming, continuous glacier retreat and ablation of differences in the debris cover have led to the formation of a large number of glacial lakes and the continuous expansion of glacial lake areas (Nie et al. 2017, Chen et al. 2021a). In the past 30 years, the number of glacial lakes in High Asia has increased by 17.4%, the total area has increased by 17.3%, and the glacial lake area in the whole region expanded by 0.58%/a (Zhang et al.2022a). The rapid change of glacial lakes may increase the possibility of the occurrence of glacial lake outburst floods (GLOFs) (Zhong et al. 2021). The risk of GLOFs in High Asia is the highest (Taylor et al.2023). This may threaten the lives and property of 30 surrounding residents, and downstream infrastructures (Song et al. 2016; Begam et al. 2018; Nie et al. 2023). Such as, the GLOF in Tibet on June 26,2020, led to the destruction of 43.9 kilometers of roads and 8 bridges, and the flooding of 19.98 hectares of farmland (Zheng et al. 2021). Therefore, continuous dynamic monitoring of glacial lakes is essential to studies on climate change, water resource distribution, and disaster warnings. However, many small and unevenly distributed glacial lakes are ignored, these glacial lakes usually have a high risk of outburst (Zhang et al. 2022b).

✧ **Comment 4:**

*Newly inserted text needs a reference (L28 to L30) - would also suggest rephrasing as increasing glacial lake sizes does not necessarily mean it will increase the occurrence of GLOFs. I would rephrase to*

*suggest that the magnitude of GLOFs could increase as glacial lakes expand. Also needs citations.*

**Respond:**

Thank you for your help. Increased glacial lakes do not necessarily lead to increased GLOFs, but they can increase the likelihood of GLOFs occurring. We will elaborate on this in the revised version and add corresponding citations.

Re-edit:

The rapid change of glacial lakes may increase the possibility of the occurrence of glacial lake outburst floods (GLOFs) (Zhong et al. 2021). The risk of GLOFs in High Asia is the highest (Taylor et al.2023). This may threaten the lives and property of 30 surrounding residents, and downstream infrastructures (Song et al. 2016; Begam et al. 2018; Nie et al. 2023). Such as, the GLOF in Tibet on June 26,2020, led to the destruction of 43.9 kilometers of roads and 8 bridges, and the flooding of 19.98 hectares of farmland (Zheng et al. 2021).

✧ **Comment 5:**

*L32 to L33: Check tenses, 'to be' does not seem to fit*

**Respond:**

Thank you for your suggestion. We will revisit our wording and make appropriate adjustments in the revised version to ensure grammatical accuracy.

Re-edit:

Such as, the GLOF in Tibet on June 26,2020, led to the destruction of 43.9 kilometers of roads and 8 bridges, and the flooding of 19.98 hectares of farmland (Zheng et al. 2021).

✧ **Comment 6:**

*L35: Why are glacial lakes on the of sensitive indicators to 'global changes'? Please clarify in the text*

**Respond:**

Thank you for your suggestion. Under the background of climate change, the area and number of glacial lakes show a trend of rapid change. In the revised version we respecify such relationships.

Re-edit:

Climate warming, continuous glacier retreat and ablation of differences in the debris cover have led to the formation of a large number of glacial lakes and the continuous expansion of glacial lake areas (Nie et al. 2017, Chen et al. 2021a). In the past 30 years, the number of glacial lakes in High Asia has increased

==by 17.4%, the total area has increased by 17.3%, and the glacial lake area in the whole region expanded by 0.58%/a (Zhang et al.2022a).==

✧ **Comment 7:**

*L56: Please reference glacial lake colours.*

**Respond:**

Thank you for your suggestion. We do not directly cite any specific reference regarding the color of glacial lakes. The colors we observe are based on the image data we downloaded. However, it is undeniable that due to the influence of the development environment of glacial lakes, the colors of glacial lakes in different regions are different in remote sensing images.

Re-edit:

==Due to different development environments, the morphology of glacial lakes may differ in remote sensing images (Zhao et al. 2018). Collecting more samples of different types is of great help to enhance the stability and universality of the model (He et al. 2021).== For the sake of increasing the diversity of the training dataset, except for the high Asia region, this study also collected some glacial lake samples from other continents.

✧ **Comment 8:**

*L123: Easy freezing? What does this mean? Clarify in text*

**Respond:**

Thank you for your suggestion. What we want to express here is that from the images we collected, we found that the water on the surface of the glacial lake in this place is easy to freeze. Of course, this may also be due to the image collection time. In order to avoid misunderstanding, we have removed this inaccurate statement.

✧ **Comment 9:**

*L144: Please can the authors clarify all the data in the Google Earth imagery? Readers may not have used this so can it please be clarified in the text rather than 'and other data'*

**Respond:**

Thank you for your suggestion. Google Image, also known as Google Earth, is a virtual globe software developed by Google. This software provides a wealth of satellite images, aerial images and GIS data. We provide the image source and imaging time in the modified version.

Re-edit:

Google Earth Images is a composite of a vast array of satellite and aerial photographs. These images are sourced from a variety of providers and platforms that are responsible for satellite launches. The primary contributors of high-resolution imagery include Maxar Technologies, the Centre National d'Etudes Spatiales (CNES) and Airbus. They provide IKONOS, QuickBird, Geoeye, WorldView, SPOT and Pleiades imagery.

**Table 1. Details of the glacial lake training dataset based on Google Earth images in this study. All copyrights Image ©Google Earth 2020.**

| Area | Image source | Image time | Data level | Number | Sample examples |
|---|---|---|---|---|---|
| Himalaya, Northern Tibet Plateau, Asia | Pleiades 1,2; SPOT 5,6,7 | 2010-2020 | Level14-18 (4.45-0.28m) | 5494 |  |
| Buenos Aires Mountains, South America | Pleiades 1,2; QuickBird | 2004-2016 | Level 17,18 (0.79m, 0.4m) | 3397 |  |
| Alaska Mountains, North America | WorldView 3 | 2013-2017 | Level 18,19 (0.28m, 0.14m) | 5519 |  |
| the Alps, Europe | Geoeye 1; WorldView 3 | 2015-2020 | Level 18,19 (0.41m, 0.21m) | 966 |  |